# A tool for modeling gene regulatory networks (GRN_modeler) and its applications to synthetic biology

Gábor Holló [ID] [✉], Jung Hun Park [ID], Emanuele Boni [ID] & Yolanda Schaerli [ID] [✉]

## Abstract

**Modeling and simulating gene regulatory networks (GRNs) is crucial for understanding biological processes, predicting system behavior, interpreting experimental data and guiding the design of synthetic systems. In synthetic biology, GRNs are fundamental to enable the design and control of complex functions. However, GRN simulations can be time-consuming and often require specialized expertise. To address this challenge, we developed GRN_modeler - a user-friendly tool with a graphical user interface that enables users without programming experience to create phenomenological models, while also offering command-line support for advanced users. GRN_modeler supports the analysis of both dynamical behaviors and spatial pattern formation. We demonstrate its versatility through several examples in synthetic biology, including the design of novel oscillator families capable of robust oscillation with an even number of nodes, complementing the classical repressilator family, which requires odd-numbered nodes. Furthermore, we showcase how GRN_modeler allowed us to develop a light-detecting biosensor in *Escherichia coli* that tracks light intensity over several days and leaves a record in the form of ring patterns in bacterial colonies.**

**Keywords** Simulation Tool; Gene Regulatory Networks; Light Biosensor; Synthetic Biology; Optogenetics
**Subject Categories** Biotechnology & Synthetic Biology; Computational Biology

## Introduction

Gene regulatory networks (GRNs) are intricate systems that regulate gene expression and ultimately determine cellular function and behavior (Ranganathan et al., 2018; Walhout et al., 2018). They consist of molecular regulators, including transcription factors, that interact with specific DNA-binding sites to control levels of mRNA and proteins.

The multidisciplinary field of synthetic biology applies engineering concepts to create novel biological components, devices, and systems, or to re-engineer existing natural systems (Perez-Carrasco et al., 2018; Tica et al., 2024). Among the most well-studied designs are feed-forward loops (FFLs), toggle switches, and oscillators. Feed-forward loops involve a gene regulating another gene, both directly and indirectly through another regulator. This architecture is commonly found in most living organisms (Alon, 2007; Mangan and Alon, 2003; Weldemichael et al., 2022) and can filter noise, produce signal-sensitive delays, create pulse-like responses or produce spatial stripe patterns (Alon, 2007; Lee et al., 2002; Milo et al., 2002; Santos-Moreno et al., 2020; Schaerli et al., 2014). The toggle switch, composed of two mutually repressive genes, enables bistability – cells can stably exist in one of two states – which is crucial in biological processes like cell fate decisions (Barbier et al., 2020; Briscoe and Small, 2015; Clark, 2017; Gardner et al., 2000; Perez-Carrasco et al., 2018; Saka and Smith, 2007). Genetic oscillators (Li and Yang, 2018), such as the repressilator (Elowitz and Leibler, 2000; Potvin-Trottier et al., 2016), generate cyclic gene expression, emulating a variety of physiological phenomena, from circadian rhythms, to cell cycle dynamics or spatiotemporal pattern formation during organisms' development (Gomez et al., 2008; Poon, 2016; Saini et al., 2019; Tsiairis and Aulehla, 2016). These synthetic circuits not only deepen our understanding of natural regulatory networks but also enable applications in biotechnology and therapeutic systems (Kumar and Hasty, 2023; Xie and Fussenegger, 2018).

Designing and testing synthetic GRNs is crucial in engineering novel biological systems (Benner and Sismour, 2005; Garner, 2021; Hanczyc, 2020). By carefully characterizing individual genetic components and integrating computational modeling into the design and engineering process, the field seeks to move beyond repeated cycles of trial and error. Therefore, as synthetic systems grow increasingly complex, computational modeling becomes vital for accurately predicting their behaviors. However, the expertise required in mathematics and programming presents a significant challenge for scientists with training focused on experimental work.

Modeling GRNs requires reaction kinetics simulations, which involve solving and analyzing ordinary differential equations (ODEs) or stochastic differential equations (SDEs) (Otero-Muras et al., 2023; Schladt et al., 2021; Sinzger-D'Angelo et al., 2024; Zechner and Koeppl, 2014, Faquir et al., 2025; Sequeiros et al., 2023). There are many software options to solve and analyze these systems. For example, the popular program XPP/XPPAUT

Department of Fundamental Microbiology, University of Lausanne, Biophore Building, 1015 Lausanne, Switzerland. ✉E-mail: gabor.hollo@unil.ch; yolanda.schaerli@unil.ch

(Ermentrout, 2002) can be used for dynamic system analysis. To facilitate the creation of ODEs based on the stoichiometry and rate laws of reactions, specialized reaction kinetics software such as KinTek Explorer (Johnson, 2009; Johnson et al., 2009a,b) are also available.

In the early 2000s, the systems biology markup language (SBML) (Hucka et al., 2003, 2004) was developed to standardize the representation of biological process models. SBML is a computer-readable format that includes information on species, reactions, parameters, rules, and constraints. Its purpose is to enable researchers to work with the same model across different software environments. As a result, several widely-used programs for modeling biochemical reaction networks support SBML, including TABASCO (Kosuri et al., 2007), COPASI (Hoops et al., 2006), Bioscrape (Pandey et al., 2023), libRoadRunner (Somogyi et al., 2015; Welsh et al., 2022), SBMLsimulator (Dörr et al., 2014), and the MATLAB SimBiology toolbox.

Another set of tools facilitates the design of synthetic systems via a model-based design strategy. For example, txtools focuses on RNA structure analysis (Garcia-Campos and Schwartz, 2024). iBioSim (integrated biological systems simulator) (Myers et al., 2009; Watanabe et al., 2019) provides an intuitive graphical interface that simplifies the construction and visualization of genetic circuits using fundamental components such as promoters, ribosome binding sites, and terminators. TX-TLsim (transcription-translation simulator) (Singhal et al., 2021) is a computational tool designed to model and simulate the transcription and translation processes in cell-free systems, such as *Escherichia coli* extracts. It offers detailed simulations of these processes, enabling researchers to predict the behavior of in vitro genetic constructs with high accuracy. On the other hand, Metabolic Tinker is used for designing metabolic pathways (McClymont and Soyer, 2013), and WebCM supports individual-based simulations of multicellular microbial populations (Philippou et al., 2024). Finally, Flapjack (Yáñez Feliú et al., 2021) concentrates on the whole design, build, test, learn cycle (DBTL).

However, when describing the qualitative behavior of complex systems with many nodes (a regulatory unit composed of genetic elements such as a promoter, a gene and a terminator) or designing circuits for specific functionalities, these tools can still be limiting. In such cases, it is more efficient to design circuits using individual nodes as fundamental building blocks, avoiding the need to repeatedly add the same reactions and species for each node in the model-building process. This higher level of abstraction provides a simplified overview of the system, enhancing the feasibility of constructing complex networks. The genetic design automation tool LOICA (logical operators for integrated cell algorithms) uses a high-level design abstraction by representing networks as combinations of components, rather than as combinations of individual genetic parts (Vidal et al., 2022). However, it only offers a command-line interface, limiting its accessibility to users who already possess modeling expertise.

We have developed a user-friendly software with a graphical user interface (GUI) specifically designed for biologists who may not have advanced computational skills. It also offers command-line support for advanced users. Our application is an intuitive and valuable tool for synthetic biology, designed specifically to model and simulate synthetic GRNs, providing fast feedback on their dynamical behavior to assist in the characterization of new circuits.

It enables users to model GRNs, predict and compare their behavior with ease. The tool simplifies the creation of novel circuits, facilitates rapid hypothesis testing, and guides the design process by highlighting where strong or weak interactions are needed to construct robust systems that are less sensitive to minor parameter changes. Additionally, it can be used to make predictions for potential experiments and genetic modifications, helping researchers anticipate results. For circuits that do not perform as expected, the application assists in troubleshooting and diagnosing issues, thereby enhancing problem-solving and optimization efforts. In summary, by simplifying the modeling process, our tool empowers researchers to design and engineer GRNs more effectively, thus accelerating progress in synthetic biology.

We demonstrate the main capabilities of the application and illustrate their use with examples. We start by modeling well-established circuits from the literature, focusing on oscillators and feed-forward loops. Oscillators have been a subject of great interest for synthetic biologists since the early days of the field (Rossi and Dunlop, 2018). Our analysis includes several oscillators: the repressilator, the first synthetic oscillator described in 2000 (Elowitz and Leibler, 2000), and an extension of it that facilitates tuning of amplitude and period (Tomazou et al., 2018). We also present a model of the CRISPRlator—a synthetic oscillator with the repressilator topology, but leveraging CRISPR interference (CRISPRi) instead of protein transcription factors to inhibit transcription (Santos-Moreno et al., 2020). In addition, we include models of feed-forward loops (FFLs) (Alon, 2007; Mangan and Alon, 2003; Weldemichael et al., 2022) and the bistable toggle switch (Gardner et al., 2000). After showcasing the capabilities of our application with known systems, we illustrate how it can be used to design novel circuits. As an example, we create new, robust oscillators that can oscillate with an even number of nodes. Finally, we present an example of the interplay with experimental work. We build a light-recording biosensor in *E. coli* and demonstrate how GRN_modeler predicts phase and anti-phase output signals in response to periodic external light exposure, such as day-night light cycles. We then validate these predictions through experimental testing, showcasing the tool's effectiveness in guiding biological circuit behavior.

## Results

### Basic properties of the application

Our application is based on MATLAB SimBiology, a powerful toolbox within the MATLAB environment designed for modeling, simulating, and analyzing biological systems. Its integration with MATLAB provides significant advantages in terms of computational capabilities, data visualization tools, and an extensive library of functions for data analysis. Furthermore, SimBiology supports the SBML format, which allows users to build models efficiently within our software and then perform further analysis using any other SBML-compatible software. However, one limitation of SimBiology is that its stochastic solver is restricted to mass action kinetics, where the reaction rate is proportional to the product of the reactant concentrations. This constraint prevents the modeling of complex reactions, such as enzymatic reactions with Michaelis–Menten kinetics, or inhibitions and activations, which are typically described using Hill functions. To address this, we

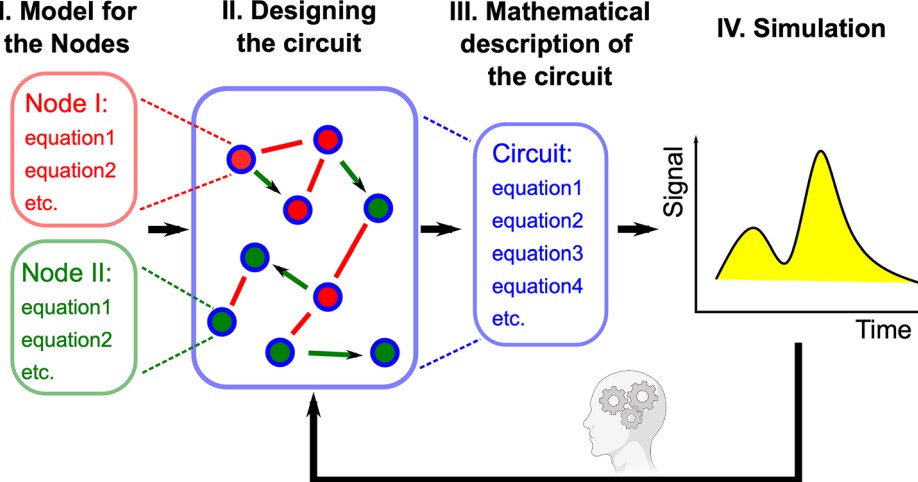

**Fig. 1. Flow chart of the application modeling steps.**

First, a model for each node must be chosen (a selection of existing models is made available in the app, or a new model can be implemented). Then, using the graphical user interface, users can assemble a circuit. Next, the application automatically generates the corresponding differential equations for the entire system, allowing simulations to be run and to observe the dynamic behavior of the GRN. If the circuit does not produce the expected results, users can redesign it. Once experimental data is available, the parameters can be fine-tuned to achieve a closer quantitative match with the simulations.

have integrated our application with COPASI (Hoops et al., 2006) through BasiCO (Basic COmpartmental model Interface) (Bergmann, 2023), a streamlined Python interface for COPASI. This integration enables access to a broader range of solvers, including stochastic ones, within our MATLAB application, leveraging COPASI's extensive solver capabilities. Most of the functionalities in our application become available after installing SimBiology. To access the COPASI solvers, it is also necessary to install COPASI (Hoops et al., 2006), Python, and BasiCO (Bergmann, 2023).

The choice between deterministic and stochastic modeling in biological systems depends on the system's scale and the role of intrinsic noise. Deterministic simulations, often using ordinary differential equations (ODEs), are suitable for large systems where molecular fluctuations are negligible, like metabolic networks. In contrast, stochastic simulations (e.g., Gillespie's algorithm) are key for systems with low copy numbers or inherent randomness, such as gene expression or signaling pathways. Hybrid models combine both approaches, using deterministic methods for fast processes and stochastic ones for slow, noise-sensitive reactions. Choosing the right framework is crucial for accurate biological representation. Our software supports both deterministic and stochastic simulations, with available solvers listed in Appendix Table S1.

The key concepts and the level of abstraction that demonstrate how the GRN_modeler operates are summarized in Fig. 1.

I.  First, we define a model for each node, what we call "node model" to distinguish from the mathematical description or the model of the whole circuit, what we will call "circuit model" or simply "model". The user can select an implemented node model, or create a new one. New node models can be included in the models folder, and these node models can be selected when starting the application. Our simulation functions at the node level, where nodes are here defined as the fundamental components of the biological network. Each node represents crucial elements such as reactions and concentrations of species (e.g., RNAs and proteins) produced through transcription and translation from the DNA coding sequence. In the node model, nodes can have varying input and output functions, allowing users to define interactions between them (Appendix Fig. S1). Kinetic parameters can be either "individual" or "common". For individual parameters, a separate parameter is assigned to each node, named accordingly. For "common" parameters, all nodes use the same value.

II. Second, once a model for a node is established, the system can be constructed by adding nodes, defining interactions between them (Appendix Fig. S2), and incorporating external regulators or protein degradation by proteases.

III. Third, the application automatically generates the corresponding differential and algebraic equations for the entire system, based on the model for the nodes and their interactions. Once the mathematical model for the circuit is constructed, we can further refine it by applying various "rules" to fine-tune the system's behavior. In SimBiology, "rules" define mathematical relationships between model components, such as differential equations, algebraic expressions, or rate laws, to govern the dynamics of biological systems. For example, inducer concentrations can be temporally modified by these user-defined rules. We will demonstrate this property in the section "Experimental example: a light biosensor" by modulating the parameters sinusoidally in response to external light exposure.

IV. In the final step, simulations are run to observe the system's dynamic behavior. The goal of our application is to accelerate circuit design and provide an easy way to visualize potential outputs. If the circuit does not exhibit the expected behavior during simulations, the design can be revisited in step II, allowing for adjustments before repeating the process. In this step, human creativity plays a crucial role. While the application assists in

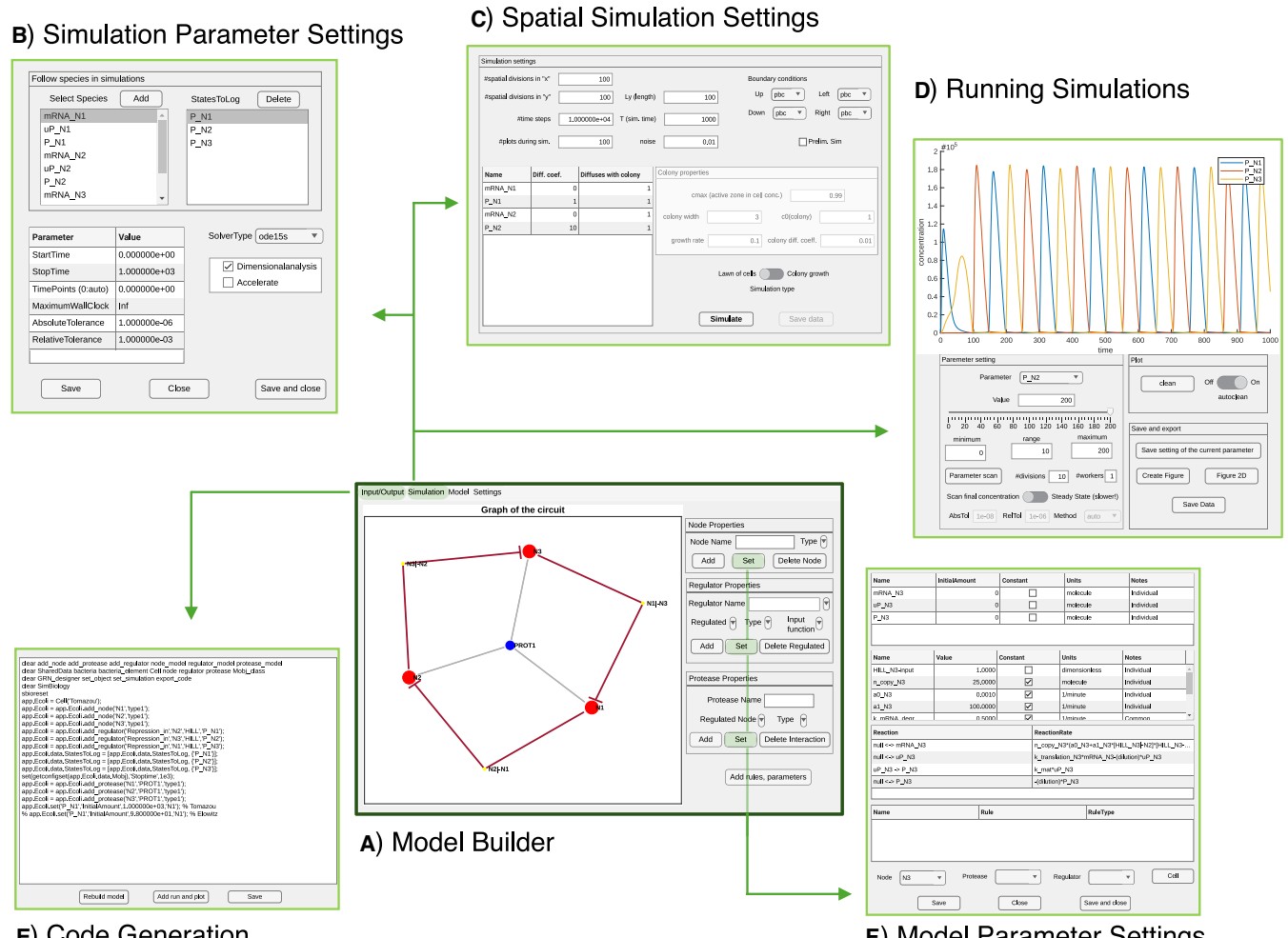

**B) Simulation Parameter Settings**

**C) Spatial Simulation Settings**

**D) Running Simulations**

**A) Model Builder**

**E) Code Generation**

**F) Model Parameter Settings**

**Fig. 2. Key components of the graphical user interface.**

(**A**) Model builder for designing models, (**B**) Simulation parameter settings for configuring simulation parameters, (**C**) Spatial simulation settings for growing colonies and lawn of cells, (**D**) Running simulations for executing and managing simulations, (**E**) Code generation for converting models into executable code, and (**F**) Model parameter settings for adjusting model-specific parameters.

predicting and visualizing circuit behavior, it is the user's knowledge and intuition that guide the necessary modifications to the design.

We designed a GUI (Fig. 2A–F) to enable users to create the model for the circuit, set properties, and run simulations without requiring any programming skills. It provides access to all the functions, while allowing users to select objects and set their parameters in an intuitive way. The constructed circuit can be visualized as a directed graph with multiple layout options, such as organizing nodes in a circular or layered structure. Users can also compare deterministic and stochastic simulation results by overlaying their trajectories and adjusting specific parameters. Additionally, the tool allows users to scan a parameter across a defined range and observe its impact on the trajectories at a specific time point. For users who prefer not to use the GUI, all functionalities of GRN_modeler can also be achieved through the command line and

the basic functions are presented in the Command Line Functionality section in the Appendix Section 1.1. While the GUI provides many useful properties and functionalities, MATLAB offers additional powerful tools for model analysis. For this reason, once the user has built the model for the circuit in the GUI, the code generation feature allows one to produce a MATLAB script that replicates all actions performed in the GUI. This approach enables users to work more conveniently without needing to know the specific species or parameter names or the syntax of the different functions. Finally, it is also possible to open the SimBiology Model Builder application or COPASI directly from the GUI for further analysis or parameter fitting.

## Building GRNs and the implemented node models

To demonstrate the capabilities of our tool in simulating GRNs, we have implemented several node models (Appendix Fig. S1). In the

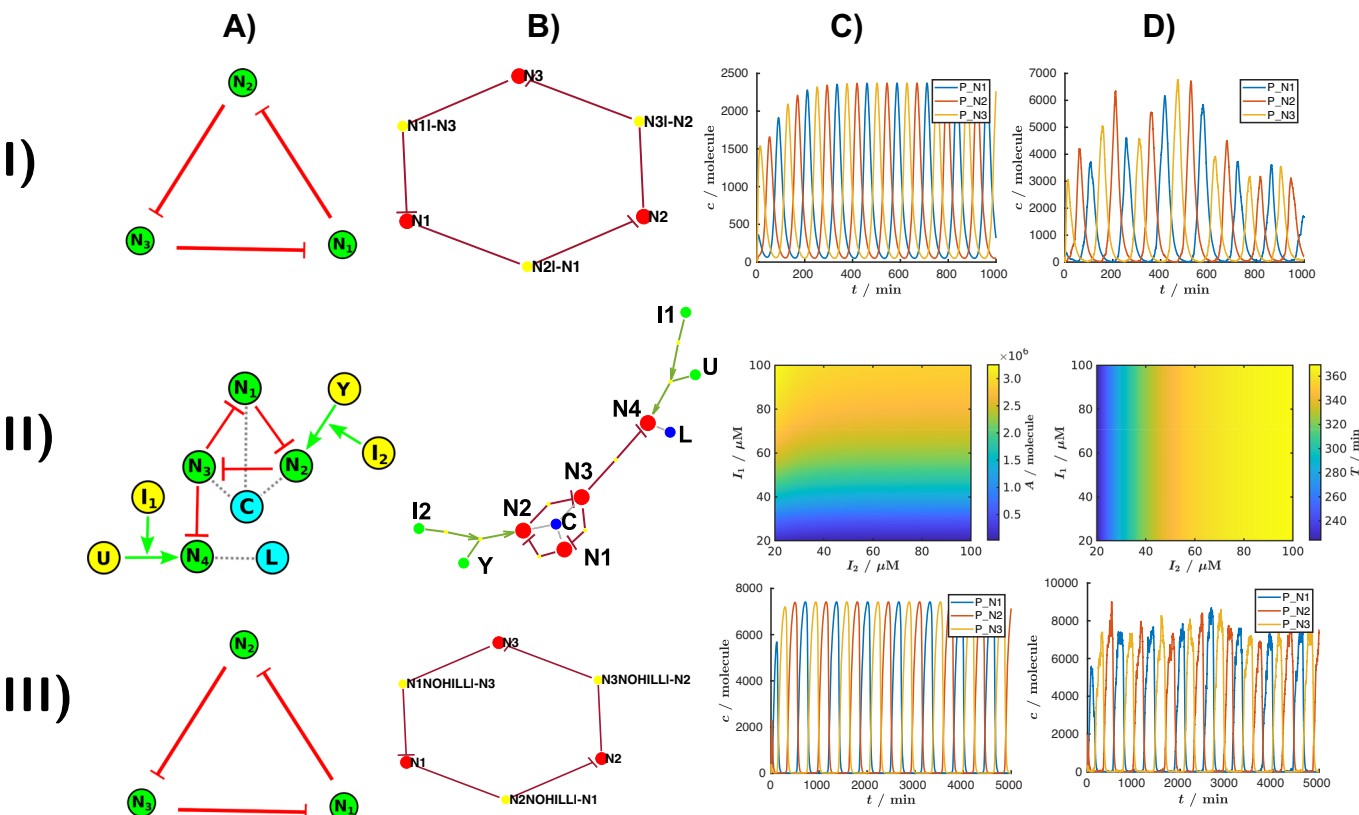

**Fig. 3. Applications of three different node models implemented in GRN_modeler.**

I) The repressilator (**A**) Topology of the repressilator. (**B**) Topology of the circuit in the interface generated by the `make_graph()` method: the red dots $N_1$, $N_2$, and $N_3$ represent the nodes in the circuit. The red arrows denote the directed edges of the graph, illustrating the repression interactions between nodes. Additionally, the yellow dots $N_1| - N_3$, $N_2| - N_1$, and $N_3| - N_2$ represent the regulatory interactions. For example, $N_1| - N_3$ denotes that node $N_1$ is repressed by $N_3$. (**C**) Deterministic and (**D**) stochastic simulations of the repressilator using the Elowitz model, as described in Appendix Table S3 in the Appendix. The detailed information about the model is available in the file "repressilator.html". II) Redesigned repressilator for an independent modulation of amplitude and frequency. (**A**) Topology of the redesigned repressilator. "C" and "L" are proteases, "$I_1$" and "$I_2$" external inducers, "U" and "Y" transcription factors activating $N_4$ and $N_2$, respectively. The dotted gray lines indicate which proteins are degraded by the proteases. (**B**) Topology of the circuit in the interface. (**C**) The effect of the inducers on the amplitude of $N_4$. (**D**) The effect of the inducers on the time period of $N_4$. The detailed information about the model is available in the file "Tomazou.html". III) The CRISPRlator circuit. (**A**) Topology of the CRISPRlator. (**B**) Topology of the circuit in the interface. For example, $N_1$NOHILL$|-N_3$ denotes that node $N_1$ is repressed by $N_3$ through an input named NOHILL, which corresponds to the CRISPRi interaction in the node model. This repression is incorporated into the system by introducing new reactions ($R_6$ and $R_7$ in Appendix Table S5), where the dCas:sgRNA$_{N3}$ complex binds to DNA$_{N1}$, inhibiting transcription at node $N_1$. (**C**) Deterministic and (**D**) stochastic simulations of the CRISPRlator generated by the model detailed in Appendix Table S5 in the Appendix. The detailed information about the model is available in the file "CRISPR.html". The deterministic simulations were fulfilled with the "ode15s" solver of MATLAB, while the stochastic simulations were performed using the "adaptivesa" solver of COPASI in every case.

following sections, we illustrate how to build and reproduce the behavior of well-known oscillators and feed-forward loops. These examples can serve as templates for creating new node models or adapting existing ones from the literature. All examples presented in the manuscript can be found in the analysing_examples folder of the code.

### The Elowitz node model

First, we implemented a node model for the repressilator. The repressilator, one of the first synthetic genetic circuits, was published by Elowitz and Leibler (2000). This circuit consists of a network of three transcription factors (TFs) that sequentially repress each other's expression in a cyclic manner, generating oscillatory behavior (Fig. 3IA,B). The repressilator showed oscillations in *E. coli*, demonstrating the feasibility of engineering periodic

gene expression in living cells. The node model describing the repressilator (referred to in this manuscript as the "Elowitz model") only includes mRNAs and transcription factors, making it an ideal choice to describe GRNs based on transcription factors. More details on the Elowitz model are provided in Appendix Table S3, as well as a detailed description of the reaction kinetics simulations generated by SimBiology. In Fig. 3I, we present the results of both deterministic (Fig. 3C) and stochastic simulations (Fig. 3D). For the stochastic simulations in this example, we utilized the "adaptivesa" solver from COPASI. Notably, the fundamental model describing the nodes (e.g., RNA, protein concentrations and the repression) is not specific to the repressilator and can be applied to simulate systems with different topologies. This generalization allows the nodes to be used as building blocks for designing diverse circuits with varying functions and purposes.

### The redesigned repressilator node model provides independent control of amplitude and period

A more refined node model can be applied to the same genetic circuit, increasing the level of detail of the simulation. Building upon the Elowitz model, Tomazou et al. (2018) developed a more complete transcription factor node model, which includes the processes of protein folding and protein degradation by proteases. We show the details of the Tomazou model in Appendix Table S4. In this node model, transcription ($R_1$) produces an (mRNA) which is translated ($R_2$), yielding first an unfolded protein (uP) which finally becomes, through maturation ($R_3$), a folded protein (P). Additionally, the proteins are degraded by proteases following Michaelis–Menten kinetics. Also, the concentrations of the mRNAs, folded and unfolded proteins decrease with a dilution rate ($k_d$). $R_4$ reaction describes the elimination of the folded protein due to dilution by cell growth and degradation by proteases. In particular, Tomazou et al. (2018) used this node model to redesign the repressilator in order to provide independent control of amplitude and period.

In the repressilator, amplitude and frequency are interdependent —typically, as frequency increases, amplitude decreases. However, for practical applications, it would be desirable to adjust these two parameters independently. To achieve this, Tomazou et al. (2018) and Zhang et al. (2022) suggested incorporating an additional output node (N4) into the repressilator topology. N4 is repressed by one of the three oscillatory nodes (labeled N3 in Fig. 3II). The expression of node 2 (N2) can be modulated with an inducer (via Y and $I_2$). In the simple repressilator, an increased expression of N2 would not only lead to an increased frequency, but also to a lower amplitude. However, in the redesigned topology, a small amount of N3 is sufficient to repress the output node N4. Therefore, increasing the frequency with an inducer "$I_2$" does not significantly affect the amplitude of the output in N4. Similarly, by modulating the concentration of another inducer, "$I_1$", it is possible to modulate the expression strength of N4 and change the amplitude in N4 oscillations. If N4 and the other nodes are degraded by the same protease, which can result in a protease "queuing effect", the change in amplitude affects the oscillator's time period. Conversely, if N4 is not coupled to the other nodes, for example, if they are targeted to different proteases (C and L) (Fig. 3II and Appendix Table S4), the amplitude can be modulated independently of the frequency. In Fig. 3II, we replicate these simulations, demonstrating the independent modulation of amplitude and frequency. Finally, to demonstrate the implementation of self-regulating nodes in GRN_modeler and provide additional examples, we implemented the model of two more oscillators, namely the Goodwin oscillator (Goodwin, 1963) (Appendix Fig. S5) and the dual-feedback oscillator (Stricker et al., 2008) (Appendix Table S6) in the Appendix, Section 2.6.

In summary, in this section, we illustrated how to use different node models, add proteases, and build more complex circuits. Using the example of the implemented node models, one can easily integrate new node models. Even with the increased complexity of the reaction system, building the circuit model remained simple and could be completed within minutes using our app.

### The CRISPRlator node model

Synthetic GRNs based on protein transcription factors are robust and effective, but they have some limitations. Engineering genetic circuits with CRISPR-based gene expression control represents a promising alternative (Santos-Moreno and Schaerli, 2020). In CRISPR interference (CRISPRi), a catalytically inactive Cas protein (typically dCas9) is targeted to specific DNA sequences with the help of a single guide RNA (sgRNA) and represses transcription by blocking the RNA polymerase. Due to its RNA-guided nature, as opposed to protein-based, CRISPRi allows for rapid and straight-forward design of a large number of highly orthogonal sgRNAs. CRISPR-based circuits also impose a low burden on host cells (as sgRNAs are only transcribed, but not translated) (Bikard et al., 2013; Chappell et al., 2015; Santos-Moreno and Schaerli, 2020; Santos-Moreno et al., 2020). Previously, in our group, we have built a synthetic oscillator based on the repressilator topology, but utilizing CRISPRi-mediated repression, termed the CRISPRlator (Santos-Moreno et al., 2020). Each node expresses an sgRNA in an operon with a fluorescent protein (mCherry, mCerulean, or mCitrine). The composite mRNA is cleaved by Csy4 (Tsai et al., 2014), and the resulting sgRNA binds to dCas9 to target the upstream region of the next node's promoter, while the rest of the mRNA is translated into a fluorescent reporter. The oscillation occurs analogously to the repressilator; however, the interactions between the nodes are different, thus requiring a new node model to describe the dynamics.

For this, we built upon the node model published by Santos-Moreno et al. (2023). The modifications are detailed in Appendix Section 2.4. Briefly, to align with existing transcription factor node models, we adapted parameters from the Elowitz model (Elowitz and Leibler, 2000) for reactions $R_1$, $R_2$, and $R_3$ (Appendix Table S5). To match the experimentally observed time period for the CRISPRlator (~10–11 h) (Santos-Moreno et al., 2020), we fine-tuned the dilution rate $k_d$. The revised node model is summarized in Appendix Table S5, and robust oscillations for the CRISPRlator are shown in Fig. 3III. This node model provides a general framework for the CRISPRi-based gene regulation, suitable for initial exploration of the behavior of these circuits. However, further refinement of parameters is required for more accurate descriptions of specific experimental systems.

In order to identify which parameters have the largest impact on the CRISPRlator's amplitude and time period in this model, we performed a global sensitivity analysis with the SimBiology toolbox of MATLAB. We tested the following parameters: "$a_1$" represents the promoter strength, "dCas" refers to dCas9 concentration, "k_P" and "d_P" are the rates for protein production and degradation, respectively; while "krds" and "kfds" correspond to the formation and degradation rates of the dCas complex, respectively; and "kfdsd" represents the formation rate of the dCas:sgRNA complex. The first-order Sobol index quantifies the influence of each parameter on its own, while the total-order Sobol index captures the combined effect of a parameter in interaction with other parameters. We found that the oscillation period is primarily influenced by promoter strength and dCas9 concentration (Fig. 4A). In contrast, the amplitude is determined largely by promoter strength and the rates of reporter protein translation and degradation. This oscillator operates at a slower pace than the repressilator, allowing protein concentrations to nearly saturate and providing sufficient time for production and degradation rates to equilibrate. Consequently, the concentration of dCas9 has minimal impact on the amplitude, as it does not directly affect these rates. Conversely, the protein kinetic parameters have little effect on the

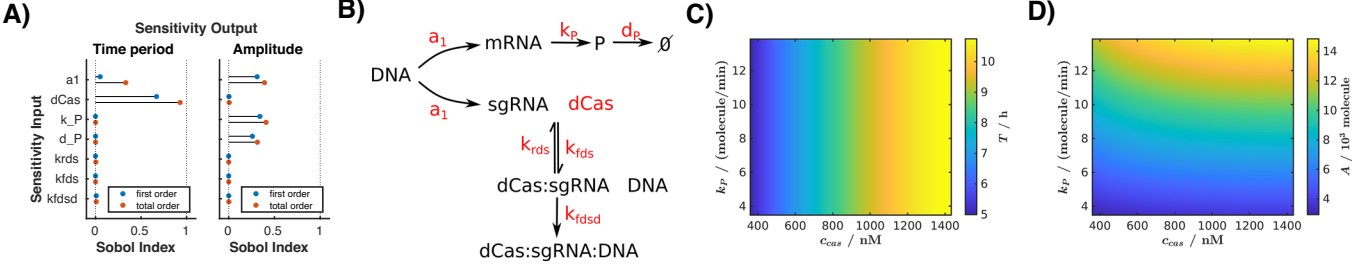

**Fig. 4. The CRISPRlator allows for independent frequency and amplitude modulation.**

(A) Global sensitivity analysis was performed using MATLAB's `sbiosobol` function, where each parameter was varied by a factor of 0.5 and two times its original value, and the number of samples was set to 1000. (B) Schematic representation of the system highlighting the examined parameters. (C) Time period $T$ and (D) amplitude $A$ of the oscillations as functions of dCas9 concentration $c_{cas}$ and protein translation rate $k_P$. The detailed information about the model is available in the file "CRISPR.html".

oscillation period, since the CRISPRlator mechanism does not depend on transcription factors, unlike the repressilator. This decoupling enables independent control of the oscillation frequency and amplitude (Fig. 4C,D) by modulating the concentration of dCas9 and the translation and degradation rate of the reporter protein, respectively. Such control produces horizontal and vertical isoclines in two-dimensional parameter space (Fig. 4B).

*The feed-forward loop network motif to showcase parameter scanning*
GRN_modeler allows easy adjustments of parameters to observe their effects on system trajectories. However, sometimes the final state of the system after a specified time is more relevant than its dynamics. For such cases, the parameter scan feature is particularly useful. This feature allows for varying a parameter over a broad range and analyzing the system's response at a specific time point. If needed, this process can be accelerated through parallelization, since these are independent simulations. This capability is integrated into the GUI.

We demonstrate the parameter scan feature using feed-forward loops (FFLs) (Alon, 2007; Mangan and Alon, 2003), a class of circuits that have attracted significant attention in SynBio (Basu et al., 2005; Greber and Fussenegger, 2010; Schaerli et al., 2014). FFLs are three-node regulatory networks where the first node controls the third node through two different paths, directly and indirectly through a second node. In coherent FFLs, the direct and indirect regulation paths have the same sign (both activations, or both repressions), while the two signs are different in incoherent FFLs. Appendix Fig. S4 shows the topologies for coherent FFLs type 1 and incoherent FFLs type 2, along with the system's response to varying $R_1$ inducer concentrations (using the parameters from the Elowitz model, Appendix Table S3). The model successfully reproduces the system's known behaviors (Alon, 2007; Mangan and Alon, 2003; Weldemichael et al., 2022): For coherent FFL 1, increasing the inducer concentration progressively activates the output node $N_3$ until it reaches saturation, as $N_3$ is activated both directly and indirectly (through $N_2$) by $N_1$. In contrast, for incoherent FFL 2, the output is activated the most at intermediate inducer concentrations, showing a peak in the response function. This behavior, known as band-pass filter or stripe-forming, has previously been implemented experimentally in SynBio (Basu et al., 2005; Greber and Fussenegger, 2010; Santos-Moreno et al., 2020; Schaerli et al., 2014).

*Identifying multistability in GRNs on the example of the toggle switch*
In most of our examples so far, the system exhibits either a single stable fixed point (e.g., the FFLs) or a stable periodic orbit (e.g., repressilator and CRISPRlator). However, many GRNs show multistability, a key feature in cell fate decisions and synthetic circuit design. In Appendix Section 2.7, we demonstrate how to identify the steady states of a multistable system using MATLAB's `sbiosteadystate` function, using the toggle switch (Barbier et al., 2020; Perez-Carrasco et al., 2018), a classic example of a bistable system, as a case study. Appendix Fig. S7 illustrates how the system transitions between stable states, depending on its initial conditions.

## Novel oscillatory circuits

In the previous sections, we implemented models described in the literature and demonstrated the capabilities of our application through these examples. However, our tool is also valuable for exploring the behavior of novel circuit topologies. It allows for qualitative testing of new circuit designs, assessing their robustness under varying parameters, and helps determine the feasibility of an experimental implementation. In this section, our goal is to illustrate how GNR_modeler facilitates the iterative process of designing new circuits, building models with the appropriate equations, and theoretically testing them.

*The reptolator*
The original repressilator has three nodes (Elowitz and Leibler, 2000). A version with five nodes has also been constructed (Niederholtmeyer et al., 2015). The 4-node repressilator does not oscillate because if every second node is activated (either N1 and N3, or N2 and N4) and the other two nodes are deactivated, the system reaches a stable state (Niederholtmeyer et al., 2015). More generally, the repressilator family can oscillate with an odd number of nodes and is bistable with an even number of nodes. We aimed to design a robust oscillator that can oscillate with an even number of nodes. By introducing mutual repressions (toggle switches) (Gardner et al., 2000) between activated nodes in a 4-node repressilator (between N1–N3 and N2–N4), this state becomes unstable, causing the system to start oscillating again, as shown in Fig. 5A,B. Since this circuit combines elements of the repressilator

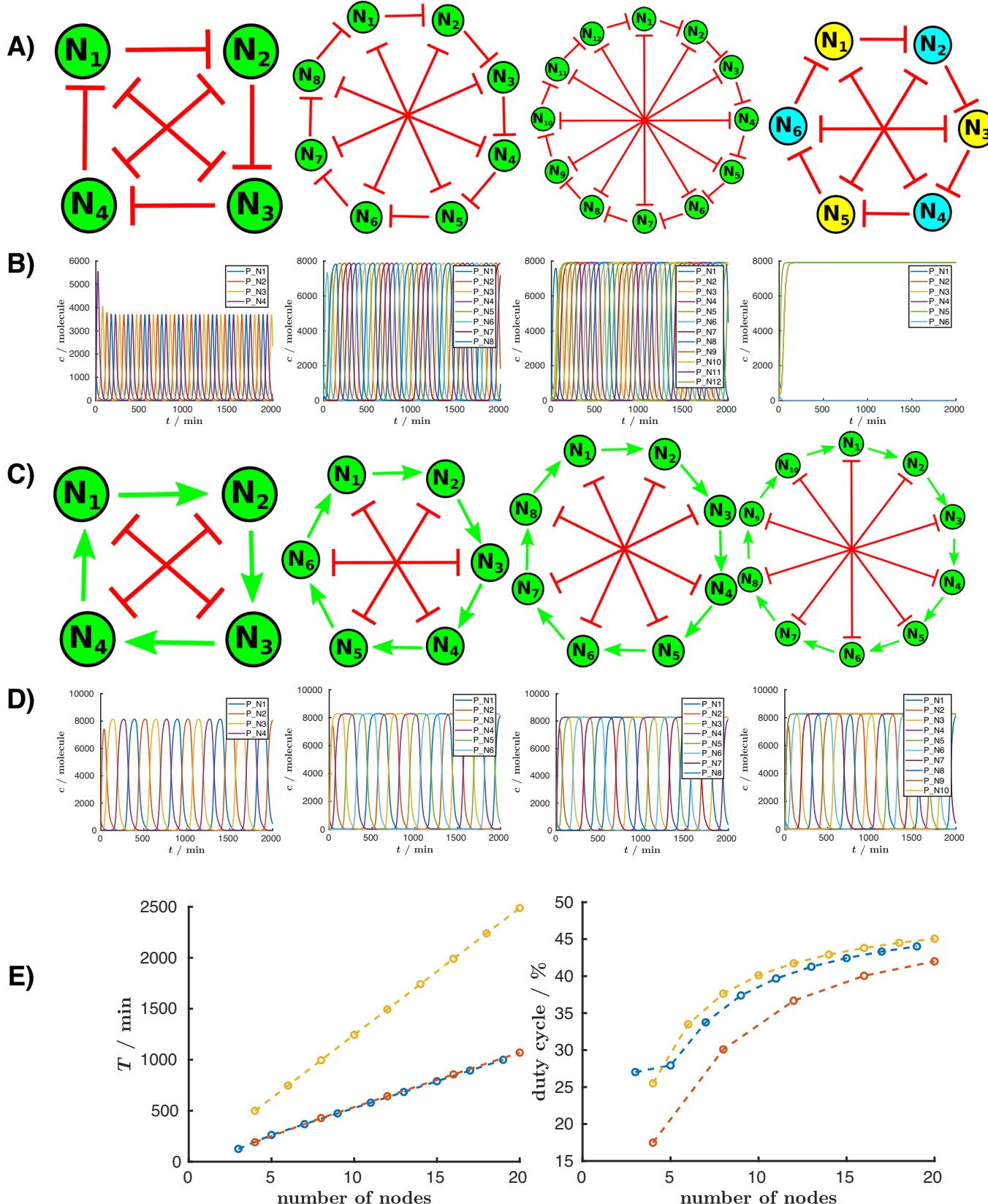

**Fig. 5. New oscillatory circuits with an even number of nodes.**

(A, B) The reptolator. (A) Topologies of gene regulatory networks with 4, 8, 12, and 6 nodes, respectively, which were created by incorporating toggle switches into the original repressilator circuit. (B) Protein concentration for each node of the circuits in (A) over time. While the circuits with 4, 8, and 12 nodes exhibit oscillatory behavior, the six-node circuit remains bistable. The detailed information about the models are available in the files "N4.html", "N8.html", "N12.html", and "N6.html", respectively. (C, D) The actolator. We designed the actolator circuit by replacing the repressions in the original repressilator with activations and introducing toggle switches between opposite nodes. (C) Topologies of gene regulatory networks with 4, 6, 8, and 10 nodes, respectively. (D) Protein concentration for each node of the circuits in (A) over time. (E) Comparison of circuit properties, specifically the oscillation period ($T$) and duty cycle (left and right, respectively), as a function of the number of nodes. The compared circuit families are: the repressilator (blue), the reptolator (red), and the actolator (yellow). The detailed information about the models are available in the files, "actolator4.html", "actolator6.html", "actolator8.html", and "actolator10.html", respectively. The deterministic simulations were performed using the Elowitz model (Elowitz and Leibler, 2000).

and toggle switches, we will refer to it as the "reptolator". Using this configuration, we can observe oscillations in systems with any integer multiple of four nodes (4, 8, 12, …$N$ $N = 4n, n \in \mathbb{Z}^+$ and $N$ is the number of nodes). Conversely, systems with 6, 10, …nodes remain bistable. Generally, if half the number of nodes is odd, this reptolator family will not oscillate. In such cases, the nodes on opposite sides of the circuit, connected by a toggle switch, can stabilize in opposite states, resulting in a bistable system, as demonstrated for the six-node circuit shown in Fig. 5A,B.

In the original repressilator, if the nodes are ordered sequentially (N1 represses N2, N2 represses N3, and so on), and the first node is activated, the sequence of activated nodes is 1, 3, 5, …, N, 2, 4, …, N-1 ($n_j = n_{i+2}$, where $i, j \in \mathbb{Z}_N^+$ and $n_i$, $n_j$ are the actual and next node while $N$ is the number of the nodes, while $\mathbb{Z}_N^+$ describe the set $\{1, 2, …, N\}$ under modulo $N$ arithmetic). In other words, every second node is activated because two consecutive repressions effectively result in an activation, causing the first node to activate the third one. In this new reptolator family, the pattern of activated nodes is more complex. Let us illustrate this with the eight-node circuit (Fig. 5A,B): If N1 is activated, the next activated node is not the third one, because it is repressed by both the second and seventh nodes. Instead, the next activated node is the sixth one because the first node represses both the second and fifth nodes. In other words, the first node releases repression of the sixth node through two pathways (N1 represses N5, which represses N6, and N1 represses N2, which represses N6), and no other node inhibits the sixth node. Following this logic, the next activated node is the third node, and the system oscillates in the following sequence: N1, N6, N3, N8, N5, N2, N7, and N4. Other circuits in this family follow the same activation rules ($n_j = n_{i+N/2+1}$, where $i, j \in \mathbb{Z}_N^+$ and $n_i$, $n_j$ are the next and previous node, while $N$ is the number of the nodes).

### The actolator

The repressilator can oscillate with an odd number of nodes, while the reptolator can oscillate with an integer multiple of four nodes. However, these circuits cannot oscillate with 6, 10, 14, … nodes ($N = 4n + 2$, where $n \in \mathbb{Z}^+$, and $N$ is the number of nodes). To create an oscillator capable of operating with 6 or 10 nodes, we designed the "actolator". This circuit is similar to the reptolator, except that the repression loop is replaced with an activation loop (Fig. 5C,D). Compared to repression, where a node is active in the absence of repression, in activation, a node is not expressed until activation occurs. Unlike the reptolator, the six-node actolator is not bistable, and this circuit oscillates for every even number of nodes. If we consider the state in which N1 levels are high and N6 levels are low, it means N2 is being activated. At the same time, N1

is not activated by any node, so it slowly gets depleted. Once N1 levels are low, N2 is no longer activated, and this instability makes the system move forward. Another difference compared to the repressilator or reptolator families is the order of activated nodes. Since the circuit is based on an activation loop, the activation of nodes follows the sequence of N1, N2, N3, … ($n_j = n_{i+1}$, where $i, j \in \mathbb{Z}_N^+$ and $n_i$, $n_j$ are the actual and next node, while $N$ is the number of nodes). It is worth noting that there is an alternative ladder-like representation of the actolator (Appendix Fig. S9C,D), which highlights the role of the toggle switches in the circuit more than the activation loop. In the four-node actolator, if node N1 is active, it will, in turn, activate node N2. Note that N1 and N2 are on two separate toggle switches, and if N2 is active, it will repress node N4, preventing it from activating the first node, so the first toggle switch will turn off. In the next iteration, the first toggle switch would turn on again, this time with a high level of N3. In other words, the two toggle switches periodically destabilize each other with a delay (due to interactions), causing them to switch one after another.

In Fig. 5E, we compare the oscillation period and duty cycle (the proportion of time during which the system is active) across three oscillator families: the repressilator, the reptolator, and the actolator. For the duty cycle calculation, a node is considered active when its protein concentration exceeds the midpoint between its minimum and maximum values during one oscillation period. The data show that as the number of nodes increases, the duty cycle approaches ≈50%. As a qualitative example, we can consider the repressilator, where each node is repressed by only one other node. If the average duty cycle of any given node were to fall below 50%, the node it regulates would subsequently exhibit a higher duty cycle. Due to the symmetrical nature of the circuit, this ultimately results in an average duty cycle of ≈50% for all nodes. However, when the number of nodes is lower, the protein concentration either does not reach saturation or takes longer to do so, resulting in a decreased duty cycle. This increase in duty cycle with increasing nodes, combined with a longer oscillation period, suggests that larger circuits could impose a significant metabolic burden on cells. If we aim to design larger oscillators in the future, it would be beneficial to develop GRNs where the duty cycle decreases as the number of nodes increases. Another noteworthy model prediction is that, for the same number of nodes, the actolator oscillates at a slower rate than the reptolator and repressilator. This difference arises because, in the longest (or rate-determining) pathway between two consecutively activated nodes, the repressilator and reptolator each require the deactivation of one intermediate node, whereas the actolator requires the consecutive deactivation of two nodes (The regulation

route for the repressilator: $n_i - |n_{i+1} - |n_{i+2}$, for the reptolator: $n_i - |n_{i+1} - |n_{i+1+N/2}$ and $n_i - |n_{i+N/2} - |n_{i+1+N/2}$, while for the actolator: $n_i \rightarrow n_{i+1}$ and $n_i - |n_{i+N/2} \rightarrow n_{i+1+N/2} - |n_{i+1}$, where $n_i$ denotes the actual activated node, $N$ is the number of the nodes and $i, j \in \mathbb{Z}_N^+$.). As a result, the predicted oscillation periods for the repressilator and reptolator are similar, while the actolator oscillates with roughly twice the period. In Appendix Fig. S8, we present the special case of the two-node actolator.

### Modified repressilator circuits: the acrelator family

As discussed previously, the repressilator with an even number of nodes is typically bistable. Instead of adding toggle switches, another way to destabilize such a system and induce oscillations is by replacing an odd number of repressive interactions with activations. We named this circuit, which integrates both *activations* and *repressions*, the "acrelator". The accelerator circuits, similarly to the repressilator, are ideal in that they contain the minimum number of regulatory connections, matching the number of nodes in the system. Fewer connections would result in some nodes remaining unregulated. An additional advantage of these circuits is the presence of a lower duty cycle in certain nodes. Further details of these circuits can be found in the Appendix Section 2.8.2, Appendix Figs. S10–S12.

In summary, by creating these novel oscillatory families, we have demonstrated how to use GRN_modeler to go from a desired behavior (oscillations with even numbers of nodes) to circuit design. Although this is not an automatic process, the tool accelerates circuit design by revealing the system's dynamic behavior. It is also worth noting the advantage of the command-line functionalities, which allow for the modular construction of large circuits. This capability enabled us to easily test the behavior of circuits with a large number of nodes. In the Appendix Section 2.1, we show that the scalability of the system is effective, as simulation time increases linearly with system size due to the efficient solvers implemented in MATLAB and COPASI, allowing for the efficient examination of large systems (Appendix Fig. S3 and Appendix Table S2).

## Experimental example: a light biosensor

In the previous section, we demonstrated how to use GRN_modeler to theoretically explore the behavior of new circuit designs. However, preliminary experimental data are often available when investigating a GRN. In this case, a model can help to gain further insight to support data interpretation and to suggest additional experiments. In this section, we provide an example demonstrating the interplay between GRN_modeler and experimental approaches in the development of a light-sensitive biosensor.

We sought to engineer *E. coli* as a bacterial light biosensor. We envisioned *E. coli* growing on a solid gel surface and tracking light intensity and fluctuations over several days. Light exposure regulates the expression levels of the biosensor circuitry, resulting in dynamic encoding of the input in the form of concentric rings within the colony. This approach translates dynamical changes of the input into a spatial pattern, which is very much reminiscent of concentric seasonal rings within tree trunks. Likewise, the biosensor would allow us to decode the level of input at each given time point without continuous output recording.

### Engineering the light biosensor

To construct the light biosensor, we chose to employ the VVD-AraC (BLADE system) (Romano et al., 2021), a protein composed of the C-terminal DNA-binding domain of the transcription factor AraC fused to a light-triggered dimerization domain (Vivid, VVD) (Schwerdtfeger and Linden, 2003). In the presence of light, VVD-AraC dimerizes and binds to the pBAD promoter to activate the expression of the downstream gene. We started by characterizing VVD-AraC in liquid culture, using cells harboring the plasmids coding for a fluorescent reporter (sfGFP) controlled by the pBAD promoter, and VVD-AraC. As expected, after exposure to varying intensities of blue light, we observed an increase in fluorescence with increasing light intensity (Fig. 6A).

We implemented a one-node circuit under the control of an external inducer (i.e., light) in the GRN_modeler interface, and set the input to be alternatively "ON" and "OFF". Based on our simulations, a single-node light-sensitive circuit should be capable of tracking changes in light intensity over the day-night cycle and recording them as ring patterns. Briefly, cells with a fluorescent reporter can form ring patterns as the colony grows on solid medium. Active cells at the colony edge are inducible by light, while inner cells enter the stationary phase, "freezing" the level of fluorescence at the corresponding space-time, which creates the rings. To test this prediction, we employed a one-node circuit using light-inducible mCitrine to dynamically sense fluctuations in light. Appendix Fig. S14 shows that at high light intensities, this simple one-node biosensor resulted in alternating bright and dark fluorescent rings, effectively registering changes in light intensity. However, the sensitivity was low, with a maximum 1.25-fold difference between the brighter and darker rings.

Thereby, we wondered whether a more complex circuit with negative feedback would improve the biosensor's dynamic range output, so we opted for the CRISPRlator, since our group was already working with this circuit. We sought to construct a CRISPRlator that is light-inducible (Fig. 6B). We modeled this circuit by starting from the CRISPRlator model. Although the CRISPRlator oscillates in liquid medium (Santos-Moreno et al., 2020), it does not oscillate under the conditions tested when grown on a solid surface. Therefore, in order to resemble our biological system as closely as possible, we tuned the parameters so that the CRISPRlator model operates outside its oscillatory regime. Compared to the model presented in Appendix Table S5, we used the parameter $k_{rdsd} = 0.7762 \, \mathrm{min}^{-1}$ as suggested by Santos-Moreno et al. (2023) for non-oscillatory circuits. Additionally, the dilution rate was reduced to $k_d = 0.005 \, \mathrm{min}^{-1}$, as cell division occurs more slowly on a solid surface (Kim et al., 2018). The model generated from GRN_modeler confirmed that by applying light pulses, the expression of the fluorescent reporters should follow the input level (Fig. 6C). In Fig. 6D, we illustrate an interesting feature of our application for such cases: the ability to generate pseudo two-dimensional images from time series data. This pattern is not the result of a spatial simulation but is instead produced by assuming a constant growth rate and a small active zone at the edge of the colony, with the image generated by "rotating" the trajectories. By using this feature, the user can know the expression level of their circuit in liquid culture, and also predict the spatial patterns that the system generates when grown on solid culture.

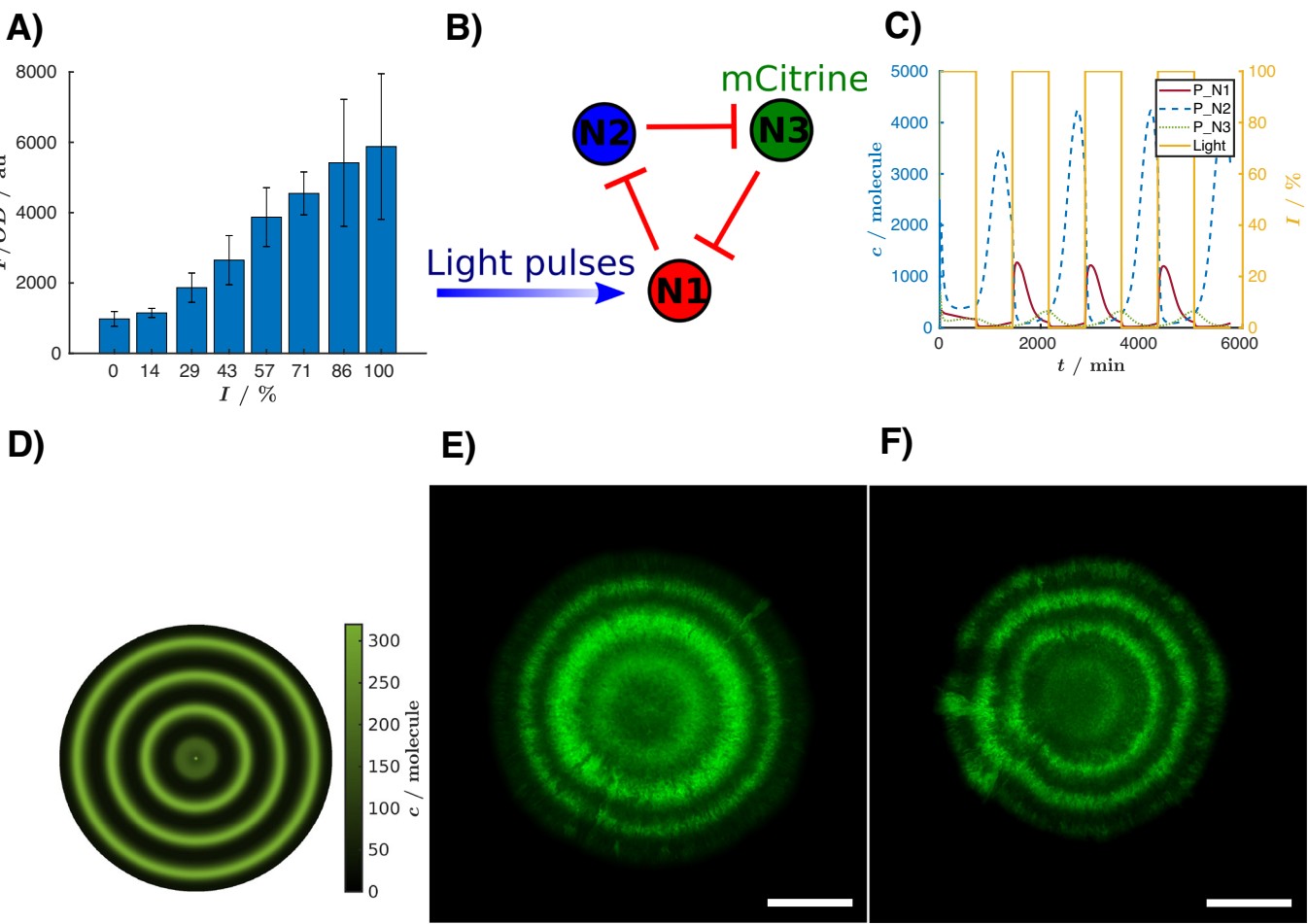

**Fig. 6. The light biosensor.**

(**A**) Characterization of the BLADE system (Romano et al, 2021). The cells express VVD-AraC, and pBAD is upstream of sfGFP. Cells were grown for 24 h at different light intensities, then culture OD and GFP fluorescence were measured by a plate reader. Bars represent mean GFP fluorescence intensity (*F, arbitrary units: au*) ± standard deviation, normalized by the optical density (OD) at 600 nm, as a function of light intensity, for three biological replicates. (**B**) Schematic diagram of the three-node light-sensitive circuit based on the CRISPRlator. (**C**) Calculated reporter concentrations and light intensity as a function of time, generated with GRN_modeler. (**D**) Pseudo two-dimensional image showing the calculated protein concentrations for node N3, generated from GRN_modeler. The detailed information about the model is available in the file "light_sensor.html". (**E**) Fluorescence image of a colony grown at room temperature and with light pulses (460 nm) for 4 days. A square wave light intensity was applied from 0 to 15% of the device's maximal intensity, with a 41.66% duty cycle (*T* = 24 h). mCitrine fluorescence is shown in green. This same image and its biological replicates (*n* = 3) are shown in Appendix Fig. S17A. (**F**) Fluorescence image of a colony grown for 4 days on the bench with direct access to sunlight. mCitrine fluorescence is shown in green. Depicted is one representative example of three biological replicates. Scale bar = 1 mm. Source data are available online for this figure.

Experimentally, we made the CRISPRlator light-inducible by cloning a pBAD promoter upstream of the mCherry node and co-transforming the circuit together with a plasmid (pJP_Bla01) coding for VVD-AraC (Fig. 6B). Similarly to the original CRISPRlator, the light-inducible version did not oscillate on solid medium (Appendix Fig. S15). We grew cells containing the light-inducible CRISPRlator on agar plates with light that changed dynamically in a square wave shape (period 24 h, 41.6% duty-cycle) —resembling day-night cycles. Light pulses led to the expression of mCitrine (shown in green) at the edge of a growing colony, resulting in concentric fluorescent rings (Fig. 6E). This proved that our light biosensor was capable of registering the changes in the light intensity. Moreover, it was able to follow the light intensity changes in a wide range of light pulses, from periods as short as 12 h to period as long as 40 h (Appendix Fig. S16). Encouraged by

these results, we demonstrated the light-sensing capabilities of our system in a real-world application: We placed the petri dish on the laboratory bench, exposing it directly to natural light. As shown in Fig. 6F, our circuit successfully detected changes in ambient light levels.

We wondered what determined the substantial difference in the performance of the one-node and the three-node circuits. As our reaction kinetics model predicted visible rings in both circuits, it seems that the observed difference arises from an effect not accounted for in these simulations. We hypothesize that cells that are initially dark and still close to the edge of the colony can still partially react to the next light pulse and increase their fluorescence signal. This leads to a low dynamic range between the brighter and darker rings in the one-node circuit. However, in the three-node circuit, there is a time delay between the light-inducible node and

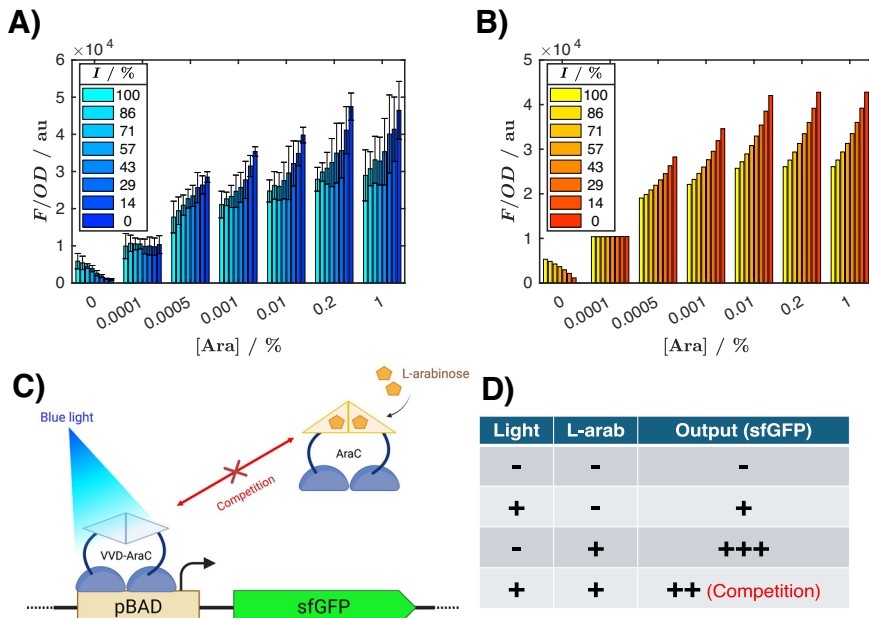

**Fig. 7.    Characterization of the system's response to light and L-arabinose.**

(A) The cells express VVD-AraC, and pBAD is upstream of sfGFP. Cells were grown for 24 h at different light intensities and L-arabinose concentrations, then culture OD and GFP fluorescence were measured by a plate reader. Bars represent mean GFP fluorescence intensity (F, *arbitrary units: au*) ± standard deviation (three biological replicates), normalized by the optical density (*OD*) at 600 nm, as a function of L-arabinose concentration and light intensity. (B) Model fitted to experimental results shown in (A). (C) Schematic illustration of the light-inducible system: a depiction of the competition between VVD-AraC and wild-type AraC, activated by blue light and L-arabinose, respectively. (D) Summarizing table showing the observed output with different combinations of inputs. Competition is seen when both light and L-arabinose are present. Source data are available online for this figure.

the reporter node, due to the time it takes for the two repressions to happen (Fig. 6B). Thanks to this time delay, the dark regions become inactive before they have time to increase their fluorescence signal. This results in a higher dynamic range between bright and dark regions, which translates to sharper rings. This phenomenon is linked to the growth of colonies, so it is not reflected in our ODE-based simulations, which consistently produce sharp rings. More detailed spatial simulations would be necessary to observe this effect computationally.

### A two-input biosensor produces anti-phase rings

Besides the plasmid-encoded VVD-AraC, our working strain also contains a genomic copy of the (wild-type) AraC transcription factor. In the presence of L-arabinose, AraC dimerizes and activates the pBAD promoter. This chemical induction is up to eightfold stronger (Fig. 7A) than the light induction. Consequently, in the presence of L-arabinose and light, both wild-type AraC (genomically expressed) and VVD-AraC (expressed from the plasmid) activate the same promoter. Contrary to our initial expectation that the two inducers would act synergistically, for L-arabinose concentrations $>10^{-4}$%, addition of light results in lower expression than induction with L-arabinose only (Fig. 7A,B). Our explanation is that VVD-AraC has significantly lower activation efficiency compared to AraC. In the presence of both inducers, VDD-AraC competes with AraC for the same promoter, thus decreasing the binding of AraC and reducing the overall expression (Fig. 7C,D). We describe this phenomenon with a two-

variable Hill function:

$$\text{HILL}([\text{Ara}], I_{light}) = \frac{([\text{Ara}]/K_{ara})^{n_{ara}} + k_{light}\left(I_{light}/K_{light}\right)^{n_{light}}}{\left(1 + ([\text{Ara}]/K_{ara})^{n_{ara}}\right) \cdot \left(1 + \left(I_{light}/K_{light}\right)^{n_{light}}\right)}, \quad (1)$$

where both light and arabinose can activate the system, [Ara] and $I_{light}$ are the arabinose concentration and the light intensity, $k_{light}$ represents the strength of the light-inducible system compared to L-arabinose, while $n_{ara}$, $K_{ara}$, $n_{light}$, and $K_{light}$ are the Hill exponents and half saturation constants for the L-arabinose and for the light-inducible system, respectively. To achieve a quantitative agreement between the model and the experimental data, we fitted the parameters (Appendix Table S6) using MATLAB's lsqnonlin function. In Appendix Fig. S13, we demonstrate how this two-input node can be handled within GRN_modeler, while Appendix Section 1.2 summarizes the different ways regulatory interactions can be interpreted with our tool.

Using the model with the fitted parameters, we simulated the patterns generated by the biosensor in response to light, in the absence (Fig. 8A) or presence of L-arabinose (Fig. 8B). The simulations predicted that the biosensor would form rings when exposed to light pulses and L-arabinose concentrations exceeding $>10^{-4}$%. Remarkably, these rings would be in anti-phase compared to the rings produced in the absence of L-arabinose (that is, dark rings would replace bright rings, and vice versa; Fig. 8A–C). We then performed the experiment with square wave light pulses in the presence of 0.2% L-arabinose. Indeed,

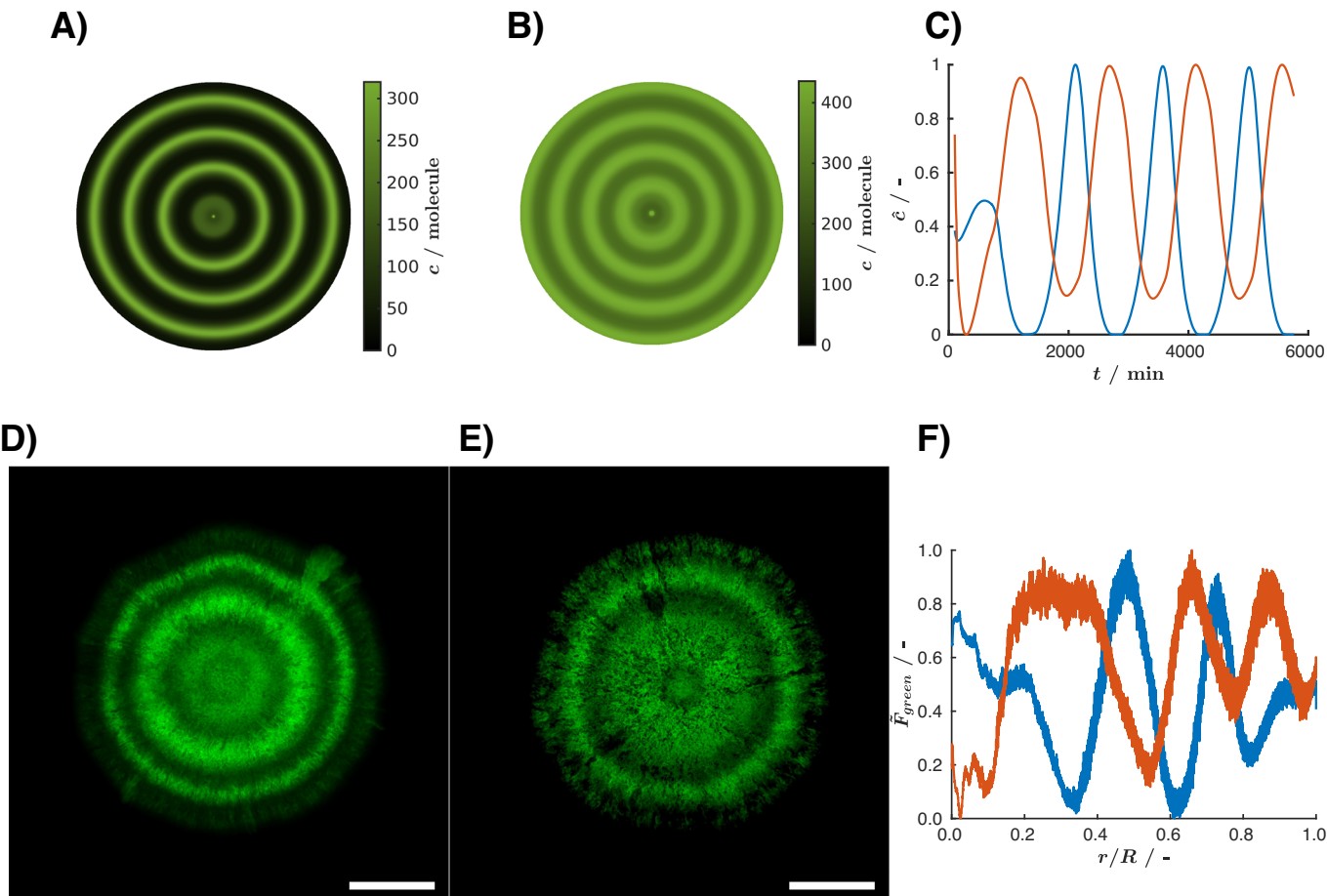

**Fig. 8. Phase and anti-phase response of the light biosensor in the absence and presence of L-arabinose.**

(A, B) Pseudo two-dimensional images based on simulations of mCitrine expression under light pulses, in the absence (A) or in the presence of 0.2% L-arabinose (B). (C) Normalized mCitrine protein concentrations over time from simulations in the absence (blue) or in the presence of 0.2% l-arabinose (red). The detailed information about the model is available in the file "light_sensor.html". (D, E) Fluorescence microscopy images showing mCitrine (green) expression in colonies harboring the light-sensing CRISPRlator. The colonies were grown under light pulses of 24 h (41.6% duty-cycle) in the absence (D) or presence of 0.2% L-arabinose (E). Scale bar = 1 mm. These same images and their biological replicates ($n = 3$) are shown in Appendix Fig. S17A, B. (F) Normalized and baseline corrected mCitrine fluorescence intensity as a function of the colony radius derived from the microscopy images of colonies (D) and (E) grown without (blue) and with L-arabinose (red), respectively. Source data are available online for this figure.

L-arabinose inverted the phase of ring patterns upon light pulses (Fig. 8D–F; Appendix Fig. S17). In other words, in the absence of L-arabinose, the light exposure leads to the expression of mCitrine (shown as green). When L-arabinose is added, the pBAD promoter is strongly activated, so in the absence of light, the entire colony is green. However, due to competition between AraC and VVD-AraC, light exposure decreases the expression of the first and third nodes (mCitrine), resulting in anti-phase rings in the colony (Fig. 8E), quantitatively consistent with the model predictions.

### Increasing the light sensitivity with L-arabinose

The use of high and low L-arabinose concentrations in the light biosensor goes beyond creating interesting patterns—it enables the development of more precise and reliable light-recording biosensors. In Fig. 9A, we illustrate how relative sensitivity to light varies with different L-arabinose concentrations under varying light

exposures. Relative sensitivity ($S_{F,I}$) is defined as:

$$S_{F,I} = \left| \frac{\partial F}{\partial I} \frac{I}{F} \right|. \tag{2}$$

This dimensionless quantity describes how the recorded fluorescence intensity ($F$) changes with variations in applied light intensity ($I$). The absolute value simplifies the comparisons across different L-arabinose concentration regions. Without this adjustment, the sensitivity at higher L-arabinose concentrations would be negative, as increased light intensity results in decreased fluorescence (producing anti-phase rings, as previously discussed). The relative sensitivity was calculated based on the function fitted in Fig. 7B. A higher sensitivity value indicates more reliable measurements. The maximum light intensity produced by the used light device (2.3 W · m⁻²) is still lower than the total sunlight spectrum (97.8 W · m⁻²) measured during the daytime in our lab, meaning that it is realistic to extrapolate the light

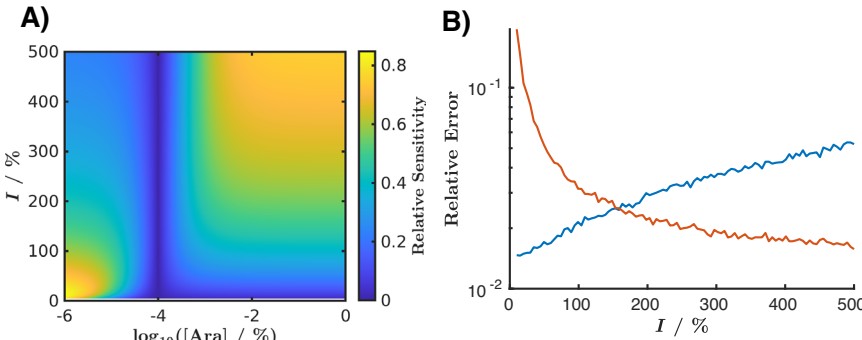

**Fig. 9.   The impact of ʟ-arabinose concentration on the light biosensor.**

**(A)** Relative sensitivity of the measured fluorescence intensity as a function of ʟ-arabinose concentration and light intensity. **(B)** Relative error from simulated experiments as a function of light intensity for low (blue curve, $10^{-6}$%) and high (red curve, $10^0$%) ʟ-arabinose concentrations. The mathematical definitions of relative sensitivity and relative error, along with details of the simulated experiments, are provided in the text.

intensity in our model to >100%. Interestingly, at low light exposure (<150%), relative sensitivity is greatest at low ʟ-arabinose concentrations (<$10^{-4}$%). Conversely, at high light intensity (>150%), sensitivity is highest at higher ʟ-arabinose concentrations (>$10^{-4}$%). For ambient light exposure, where light intensity fluctuates over a wide range, we thus predict that using two biosensors with both high and low ʟ-arabinose concentrations would ensure higher accuracy across the full intensity spectrum. Notably, the relative sensitivity approaches zero around $10^{-4}$% ʟ-arabinose, where the strength of the light and ʟ-arabinose systems are balanced, resulting in a light-insensitive system.

In Fig. 9B, we show the results of a simulated experiment investigating how arabinose concentration affects the precision of light measurements. Fluorescence intensity ($F$) was calculated for two L-arabinose concentrations: low ($10^{-6}$%) and high (1%), as a function of light intensity (ranging from 0% to 500%). To simulate experimental data ($F_{exp}$), Gaussian noise ($\mathcal{N}(0,1)$) with an amplitude $A = 0.1$ was added, resulting in $F_{exp} = (A \cdot \mathcal{N}(0,1) + 1)F$. This process was repeated 1000 times, and the light intensity was estimated from the simulated fluorescence data using MATLAB's `nlinfit` function. Finally, we determined the relative error ($err$) from the 95% confidence interval ($w_{95\%}$) of the fit:

$$err = \frac{w_{95\%}}{I_{fit}}, \qquad (3)$$

where $I_{fit}$ is the fitted light intensity. Consistent with our earlier conclusions, low ʟ-arabinose concentrations (shown in blue) yield low relative error at low light exposures, while higher ʟ-arabinose concentrations (shown in red) are advantageous at higher light exposures (Fig. 9B). Thus, by using ʟ-arabinose as an input, we can extend the operational range of our light biosensor, enabling more reliable measurements across a broader light spectrum.

## Spatial simulations

In the previous section, we demonstrated how a quick approximation of colony appearance can be generated by transforming time series data into a pseudo-2D image. However, this is a rough

estimation. With GRN_modeler, we can also perform fully two-dimensional spatial simulations in both growing colonies and static cell lawns.

For colony growth, we use the Fisher-KPP (Kolmogorov, Petrovsky, Piskunov) equation (Fisher, 1937):

$$\frac{\partial c}{\partial t} = D_c \nabla^2 c + kc(1 - c), \qquad (4)$$

where $c$ is the dimensionless cell concentration (normalized by the carrying capacity), $D_c$ is the cell diffusion coefficient, and $k$ is the logistic growth rate. As cells move, their diffusion also affects intracellular species, described by (Cao et al., 2016; Park et al., 2024):

$$\frac{\partial a_i}{\partial t} = D_c \nabla a_i \frac{\nabla c}{c} + R_i(\vec{a}, t) \cdot \Theta(c^* - c), \qquad (5)$$

where $a_i$ is the concentration, and $R_i(\vec{a}, t)$ represents the reactions of the intracellular species $i$. $\Theta$ denotes the Heaviside step function, and this term represents that the reactions cease in 'old cells' once the concentration exceeds a threshold value $c^*$, thereby restricting active intracellular reactions to the edge of the colony. A detailed derivation of the intracellular diffusion term can be found in Appendix Section 1.3.3. In GRN_modeler, users can specify which species are intracellular and affected by colony diffusion.

We demonstrate this feature using a light-inducible repressilator exposed to a sinusoidal light source (Fig. 10A). When the light period is twice the natural oscillation period, the system exhibits period-2 oscillations, resulting in alternating high-low fluorescence peaks (Fig. 10B), a phenomenon previously validated experimentally (Park et al., 2024).

Alternatively, GRN_modeler can simulate a lawn of cells, where cells are uniformly distributed on the surface of a gel. Here, two types of species exist: (i) Intracellular species, which do not diffuse and remain localized. (ii) Diffusible molecules, which mediate cell-cell interactions and may generate spatial patterns. In this case, the concentration dynamics follow a reaction-diffusion equation:

$$\frac{\partial a_i}{\partial t} = D_i \nabla^2 a_i + R_i(\vec{a}, t), \qquad (6)$$

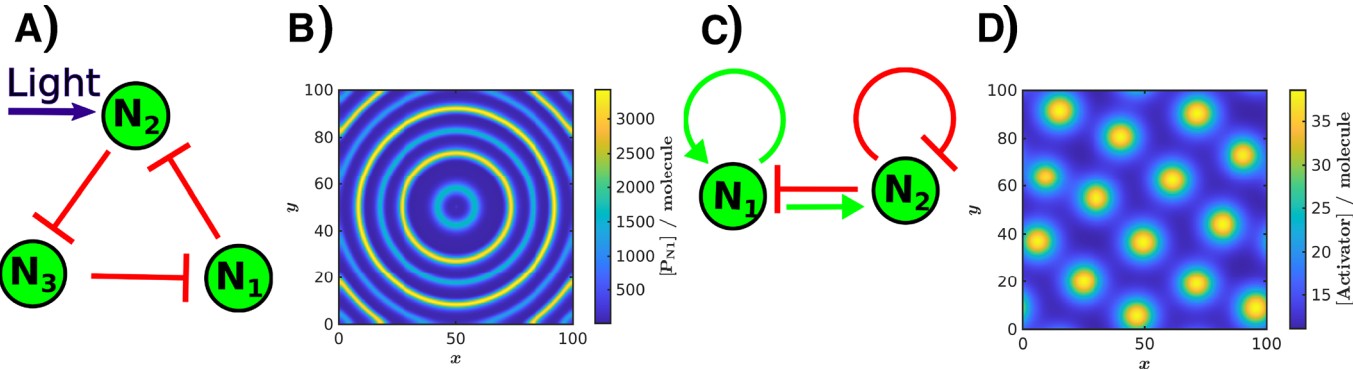

**Fig. 10.   Spatial simulations in GRN_modeler.**

(**A**) Circuit diagram of a light-sensitive repressilator. (**B**) Period-2 oscillations produced by cells expressing the light-sensitive repressilator in a growing colony, simulated with a colony growth rate of $k = 0.1$ min$^{-1}$ and a cell diffusion coefficient of $D_c = 0.01$ min$^{-1}$ (using dimensionless spatial variables). The time period of the light was 256 min and $c^{\cdot} = 0.99$. (**C**) A simple gene circuit consisting of an activator ($N_1$) and a deactivator ($N_2$), where the activator promotes both its own expression and that of the opposing node, while the repressor inhibits both its own expression and the opposing node. (**D**) Simulation of a Turing pattern in a lawn of cells containing the network in shown in (**A**). Further details of the simulation setup can be found in Appendix Section 1.3.2. The detailed information about the reaction kinetics model is available in the files: "repressilator_light.html" and "Turing.html".

where $D_i$ is the diffusion coefficient, and $R_i(\overrightarrow{a}, t)$ represents the reactions of the $i$th species.

To illustrate this, we simulate Turing pattern formation where a slow-diffusing local activator and a fast-diffusing global inhibitor interact (Dúzs et al., 2023; Menou et al., 2023) (Fig. 10C,D). Here we demonstrate how GRN_modeler can serve as a valuable tool for testing such circuits. Further details on the numerical methods applied can be found in Appendix Section 1.3.1.

## Discussion

In this work, we have developed a user-friendly application with a graphical interface for simulating GRNs (Fig. 2). Initially, we demonstrated the tool's basic features using well-established circuits. We showed how to construct the classic repressilator circuit using both the Elowitz (Elowitz and Leibler, 2000) and Tomazou models (Tomazou et al., 2018), and how to conduct both deterministic and stochastic simulations (Fig. 3). Recognizing the increasing importance of CRISPR-based circuits in synthetic biology, we incorporated and enhanced a recently published model (Santos-Moreno et al., 2023) to describe the CRISPRlator (Santos-Moreno et al., 2020) (Fig. 4). This advancement facilitates the design of subsequent CRISPRi-based genetic circuits by users. Additionally, we showcased the parameter scan functionality with examples of FFLs, illustrating the tool's capability to explore and analyze different parameter settings. We further illustrated how to examine basic dynamical behaviors—such as multistability—by identifying steady states in the example of a toggle switch, and included additional examples to show how self-regulation can be incorporated using our tool.

After presenting well-known circuits, we aimed to demonstrate how our tool can be utilized to design and analyze new GRNs. While the original repressilator family can oscillate with an odd number of nodes, we have developed two new oscillator families: the "reptolator" and the "actolator", which can give rise to robust oscillations with an even number of nodes. These designs

incorporate consecutive repression and activation rings combined with toggle switches. Furthermore, we demonstrated that by replacing some repressions with activations in the repressilator, we can create the "acrelator", which can also oscillate with an even number of nodes (Fig. 5). Furthermore, the varying duty cycles of the nodes in this configuration could be advantageous for decreasing the burden on the cells or for specific applications such as synthesizing block copolymers, where the properties are influenced by the length of the polymer blocks (Dai et al., 2004; Lee et al., 2024). The complexity of these newly designed circuits represents an exciting and challenging opportunity for their construction and experimental validation in the future. These examples highlight the value of our application as a powerful tool for streamlining the modeling process and significantly accelerating the design of new GRNs.

Finally, we presented an experimental example (Fig. 6) featuring a CRISPRi-based circuit sensitive to both light and L-arabinose (Fig. 7), demonstrating the capability of our software to predict complex dynamics. Specifically, we used the application to model competitive effects between inputs at varying concentrations, leading to phase and anti-phase responses to periodic external light signals (Fig. 8). As a potential application, colonies with this circuit could be used to track day-night light intensity changes over time. Moreover, according to the model and our preliminary experimental results, we propose that adjusting L-arabinose concentrations could enhance the accuracy of light measurements (Fig. 9). As first approximation, we visualized the formation of rings using a simple and efficient method based on reaction kinetics simulations. Furthermore, we demonstrated how we can use GRN_modeler to produce spatial simulations in growing colonies and cell lawns (Fig. 10).

As research advances, specialized tools continue to emerge for specific fields. For instance, the BioSwitch toolbox is designed for studying the dynamic behavior of gene regulatory networks (Yordanov et al., 2019), iBioSim (Myers et al., 2009; Watanabe et al., 2019), and Flapjack (Yáñez Feliú et al., 2021) allow for the construction of GRNs using fundamental components. Although

these are powerful applications, they may not be as efficient for modeling the phenomenological behavior of larger systems, while other otherwise great tools lack a GUI (Vidal et al., 2022), which can simplify the model-building process. Our interface addresses this gap by utilizing nodes as basic building blocks. This results in several advantages: it can significantly facilitate the modeling process, enhance the analysis of complex systems and provide a user-friendly GUI. It can find the steady states of a system, and furthermore, it supports spatial simulations on a growing colony or a lawn of cells. Our intuitive tool enables users to model GRNs and predict and compare their dynamical behavior with ease, accelerating the achievement of desired outcomes. Additionally, it simplifies the creation of novel circuits and facilitates the rapid testing of hypotheses, making it easier to explore and validate innovative genetic designs.

Although many excellent tools exist for simulating GRNs—such as LOICA (Vidal et al., 2022), which offers command-line functionalities, GeneEX (Kohar et al., 2020), a user-friendly web-based platform, or LimbNET (preprint: Matyjaszkiewicz and Sharpe, 2024), for spatial simulations in limb development— our application stands out in several important ways. It provides a GUI, making it accessible to users without programming experience, while also supporting command-line use. This additional functionality is particularly valuable for building larger systems or performing extended analyses not supported by the GUI, significantly increasing the flexibility of the tool.

Additionally, our GUI can generate scripts to replicate the circuit designs, further enhancing the convenience and power of the command-line interface. Beyond circuit design and simulation, our tool also enables spatial simulations, including growing colonies and cell lawns. We demonstrate this capability through examples such as non-autonomous period-2 oscillations and Turing pattern formation. We believe that this combination of usability, flexibility, and advanced features makes our application a valuable addition to the current suite of GRN modeling tools.

Using our application becomes straightforward once a model has been implemented. However, a basic familiarity with SimBiology syntax is necessary to effectively develop or implement new models. In the future, we aim to expand our application to support the implementation of node models. For instance, our current model for the CRISPR system does not yet include a description for CRISPR activation. Among others, future work will involve creating a model that incorporates CRISPR activation and fine-tuning the parameters based on experimental data.

Another limitation is that the user must manually define the circuits to be tested, as the current version of GRN_modeler does not support automatic generation of network designs that produce a desired behavior. Addressing this challenge requires a different set of approaches, and various strategies have been explored in the literature, including dedicated design toolboxes (Otero-Muras et al., 2016), brute-force screening methods (Cotterell and Sharpe, 2010), and machine learning-based frameworks (Palacios et al., 2025).

Our decision to use MATLAB in this study reflects a careful consideration of its strengths and limitations. While we acknowledge that MATLAB is a commercial platform, its multi-platform compatibility (Windows, macOS, Linux) and widespread adoption in academia ensure accessibility for our target audience. The software provides a robust environment for numerical simulations, leveraging

highly optimized solvers such as ode15s for stiff systems and SUNDIALS CVODE for large-scale differential equations. Additionally, the `sbioaccelerate` function offers significant speedup, which is especially valuable for computationally intensive tasks. In the spatial simulations, we also take advantage of MATLAB's highly optimized matrix operations—such as matrix-vector multiplication, LU decomposition, and solving linear systems. These tools enable efficient handling of computationally intensive tasks, complemented by the parallel computing toolbox for accelerated parameter sweeps. Although MATLAB's licensing cost may pose a barrier for some users, its long-term stability and backward compatibility reduce maintenance burdens compared to open-source alternatives, which often require frequent updates to maintain functionality.

Furthermore, the *SimBiology toolbox* in MATLAB provides numerous powerful features, including an intuitive graphical interface, which we have extended by integrating it with the free *COPASI* software for added flexibility. In addition to its graphical user interface, our application offers programmatic tools that are particularly beneficial for advanced users. These users can build models programmatically and perform further analyses, such as finding fixed points, evaluating the Jacobian for system stability, or running spatial simulations for growing colonies, leveraging MATLAB's extensive capabilities. In summary, the choice of MATLAB allows us to provide a reliable, feature-rich platform with high computational performance and long-term stability.

While our primary focus has been on GRNs for synthetic biology, other biologists might also use our tool to explore the properties of their favorite (naturally occurring) GRNs. Here, we have been using GRN_modeler for describing intercellular processes and GRNs. However, the framework we have developed could be adapted for other applications. In this broader context, "nodes" might represent different cells or even organs or organisms, and "regulators" could denote various interactions between these entities. With appropriate modifications and new models, our tool has the potential to model a wide range of systems.

In conclusion, we aim, with GRN_modeler, to empower (synthetic) biologists, even those with limited expertise in mathematics or programming, to effectively model their systems of interest. By bridging computational and experimental biology, this tool has the potential to accelerate scientific progress and innovation in the field.

## Methods

**Reagents and tools table**

| Reagent/resource | Reference or source | Identifier or catalog number |
|---|---|---|
| **Experimental models** | | |
| *Escherichia coli* (MK01) | Addgene | #195090 |
| **Recombinant DNA** | | |
| pJ1996_v2 | Addgene | #140664 |
| pBLADE ONLY C | Addgene | #168050 |
| pCOLA-AraC-pBAD-GFP | Duarte et al, 2017 | |

| Reagent/resource | Reference or source | Identifier or catalog number |
|---|---|---|
| pJ2072 (1-OS2) | Santos-Moreno et al, 2023 | |
| pJP_1node | This work | Addgene (#230984) |
| pJP_Bla01 | This work | Addgene (#230982) |
| pJP_Ctrl04 | This work | Addgene (#230983) |
| pJP_Osc05 | This work | Addgene (#229040) |
| **Antibodies** | | |
| **Oligonucleotides and other sequence-based reagents** | | |
| **Chemicals, enzymes and other reagents** | | |
| NEBuilder® HiFi DNA Assembly | New England Biolabs | Cat # E2621S |
| EZ Rich Defined Medium | Teknova | Cat # M2105 |
| L-arabinose | Sigma-Aldrich | Cat #3256-100G |
| Kanamycin sulfate | PanReac AppliChem | Cat #A1493 |
| Gentamicin sulfate | PanReac AppliChem | Cat #A1492 |
| Spectinomycin Dihydrochloride 5-hydrate | PanReac AppliChem | Cat #A3834 |
| **Software** | | |
| MATLAB (2023b) | https://ch.mathworks.com/ | |
| SimBiology Toolbox | https://ch.mathworks.com/products/simbiology.html | |
| COPASI | https://copasi.org/ | |
| Python | https://www.python.org/ | |
| Inkscape | https://inkscape.org/ | |
| **Other** | | |
| LITOS | *Höhener et al., 2022* | |
| Microplate reader | Synergy H1 (Biotek) | |
| Fluorescence microscope | AxioImager M1 (Zeiss) | |

## Strains and plasmids

For cloning, we used the *E. coli* strain DH5α. For the light biosensor, we used the *E. coli* strain MK01 (Kogenaru and Tans, 2014), a kind gift from Sander Tans (Addgene #195090). The plasmids were constructed using the methods described previously (Park et al., 2024). All plasmids are summarized in Appendix Table S7. Plasmids, and their maps and sequences are available through Addgene.

The light-inducible system is based on the pBLADE_ONLY_C plasmid (Romano et al., 2021). pBLADE_ONLY_C was a kind gift from Barbara Di Ventura (Addgene plasmid #168050). We modified it to contain a gentamicin resistance gene instead of the chloramphenicol resistance and deleted the f1 origin, generating plasmid pJP_Bla01. We created a reporter plasmid starting from plasmid pCOLA-AraC-pBAD-GFP (Duarte et al., 2017), which contains a ColA origin of replication, a copy of *araC* and sfGFP

under control of the pBAD promoter. By deleting *araC* from the original backbone, we generated pJP_Ctrl04.

For the one-node biosensor, we constructed plasmid pJP_1node. It contains a pBAD promoter upstream of mCitrine. It was constructed by substituting the sfGFP reporter from pJP_Ctrl04 with mCitrine, making it comparable to the three-node circuit in the ring pattern experiments.

For the three-node light- and L-arabinose-inducible biosensor, we created plasmid pJP_Osc05. It is derived from pJ2072 (1-OS2), an L-arabinose inducible CRISPRlator (Santos-Moreno et al., 2023), which contains a pBAD promoter upstream of the mCherry node. Additionally, we moved the kanamycin resistance gene between the mCitrine and mCerulean nodes. This improved the construct stability, as recombination between mCitrine and mCerulean genes would result in the loss of the resistance gene. We also removed the *araC* gene.

## Characterization of light and L-arabinose induction in liquid culture

For characterizing the light and L-arabinose sensing, we co-transformed pJP_Bla01 and pJP_Ctrl04 into MK01. A single colony was inoculated into a tube containing 3 mL of LB liquid medium supplemented with gentamicin (25 µg/mL) and kanamycin (50 µg/mL) and incubated overnight at 30 °C with shaking. 0.5 mL of the culture was pelleted, washed with saline water (0.9%) and resuspended in 0.5 mL of EZ medium (Teknova), supplemented with 0.4% glycerol. The resuspended culture was diluted 50x in EZ medium containing the necessary antibiotics and L-arabinose (final concentration of 0, 0.0001, 0.0005, 0.001, 0.01, 0.2, and 1%). For each condition, 200 µL were added to a 96-well plate (Greiner, REF #655090). The 96-well plate was placed on top of a LITOS light device (Höhener et al., 2022) and incubated at room temperature under agitation at 450 rpm (Titramax 1000, Heidolph). For every L-arabinose concentration, cells were induced with constant blue light (460 nm), at 0, 14, 29, 43, 57, 71, 83, and 100% intensity. After 24 hours, OD 600 and sfGFP fluorescence were measured (Ex: 479/12.5, Em: 520/12.5) with a microplate reader (Synergy H1, BioTek). This experiment was performed three times in parallel, corresponding to three biological replicates (for which we calculated the mean and standard deviation). For each well, sfGFP measurement was normalized by its respective OD 600. No background subtraction was performed, and the experiment was independently replicated one more time in the laboratory.

## Spatial patterning experiment

For the one-node light biosensor, we transformed pJP_Bla01, pJP_1node, and pJ1996_v2 into MK01. For the three-node light biosensor, we transformed pJP_Osc05, pJP_Bla01, and pJ1996_v2 (Addgene plasmid #140664) (Santos-Moreno et al., 2020) into MK01. Tubes containing 5 ml LB liquid medium with gentamicin (25 µg/mL), kanamycin (50 µg/mL), and spectinomycin (50 µg/mL) were inoculated with single colonies and incubated overnight at 30 °C with shaking. The cultures were serially diluted in saline water (0.9%) in order to seed ~20 colonies per plate (90 mm diameter Petri dish containing LB agar, antibiotics as indicated above and 0 or 0.2% L-arabinose). The plates were covered in aluminum foil and placed in a 30 °C incubator. After 20 h, the

aluminum foil was removed and the plates were placed upside down on top of the LITOS device (Höhener et al., 2022) for light induction at room temperature. For the phase and anti-phase experiment using the three-node circuit, light intensity varied from 0 to 15%, in a square wave shape (T = 24 h) and 41.66% duty cycle, while for the experiment with different periods of light pulses, the duty cycle was 50%. For the one-node circuit, cells were subject to light pulses with square wave shape (T = 24 h) with intensity varying from 0 to L% (L = 0, 25, 50, 75, and 100%) and 50% duty cycle. After 96 h, we took pictures of the colonies using an AxioImager M1 fluorescence microscope (Zeiss) with an Achromat 2.5X/0.12 Fluar objective. The YFP (Ex: 500/20, Em: 535/30) filter was used to measure the fluorescence of mCitrine. The exposure time was set to 100 ms.

For better visualization of the anti-phase rings, the contrast of the images was enhanced. Quantification was done on original data and not on the images with enhanced contrast.

Daylight (15-h daylength) sensing was done following the same method described above for the ring patterns without L-arabinose, but instead of placing the plates on the LITOS device, they were placed near the window in our laboratory, for 4 days, at 22 °C. The indoor white light was turned off after 18h00 and turned on from 9h00.

The experiment was independently replicated one time in the laboratory. No blinding was performed, as it was not deemed necessary given the nature of the experimental design and the objective criteria used for data acquisition and analysis.

## Image analysis

In our image analysis, we first identified the center of the colony. An intensity threshold was determined using Otsu's method (Otsu, 1979), and the colony center was located in the resulting binary image using MATLAB's *regionprops* function. Next, we transformed the image to polar coordinates and calculated the fluorescent intensity as a function of radius. Baseline correction was then applied by fitting a third-order polynomial to the intensity data and subtracting it to enhance the visibility of individual peaks. We subsequently applied a moving average smoothing with a window of 100 data points. Finally, we rescaled the intensity to the [0,1] range for better comparison across images. Figure 8F shows data from a single representative colony per condition. Additional replicates yielded consistent results but are omitted for clarity.

## GRN_modeler and simulations

We conducted deterministic simulations using MATLAB's `ode15s` and `sundials` solvers with the SimBiology toolbox. For stochastic simulations, we employed the `adaptivesa` solver from COPASI. We built the reaction networks and the differential equations using our application, the GRN_modeler. A GRN_modeler user manual for the GUI is included in the Appendix Section 1.4, and video tutorials (Movies EV1–EV8) are available as part of the Supplementary Material. All examples can be found in the analyzing_examples folder of the application. Details on initial conditions, reactions, and rate laws were generated using the SimBiology toolbox and are provided in the ".html" files. The specific filenames for these files are listed in the figure captions that display the respective simulation results.

## Data availability

The datasets and computer code produced in this study are available in the following databases: GRN_modeler software, models and examples are available on GitHub https://github.com/SchaerliLab/GRN_modeler.

The source data of this paper are collected in the following database record: biostudies:S-SCDT-10_1038-S44320-025-00148-8.

## Peer review information

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

## Acknowledgements

This work was funded by a Swiss National Science Foundation grant (310030_200532 awarded to YS), a fellowship of the Agassiz foundation (awarded to JHP), a UNIL FBM PhD fellowship in Life Sciences (awarded to JHP) and the University of Lausanne.

## Author contributions

**Gábor Holló:** Conceptualization; Data curation; Software; Formal analysis; Investigation; Visualization; Methodology; Writing—original draft; Project administration; Writing—review and editing. **Jung Hun Park:** Conceptualization; Formal analysis; Funding acquisition; Investigation; Visualization; Writing—original draft; Writing—review and editing. **Emanuele Boni:** Validation; Visualization; Writing—review and editing. **Yolanda Schaerli:** Conceptualization; Supervision; Funding acquisition; Project administration; Writing—review and editing.

## Disclosure and competing interests statement

The authors declare no competing interests.

