## [Peer Review File · Molecular Systems Biology]

A Tool for Modeling Gene Regulatory Networks (GRN_modeler) and its Applications to Synthetic Biology

Gábor Holló, Jung Park, Emanuele Boni, and Yolanda Schaerli

Corresponding author(s): Yolanda Schaerli (yolanda.schaerli@unil.ch) , Gábor Holló (gabor.hollo@unil.ch)

Review Timeline:

Submission Date:	28th Jan 25
Editorial Decision:	21st Mar 25
Revision Received:	21st May 25
Editorial Decision:	11th Aug 25
Revision Received:	25th Aug 25
Accepted:	1st Sep 25

Editor: Poonam Bheda

Transaction Report:

21st Mar 2025

Manuscript Number: MSB-2025-12880

Title: A Tool for Modeling Gene Regulatory Networks (GRN_modeler) and its Applications to Synthetic Biology

Dear Prof Schaeerli,

Thank you again for submitting your work to Molecular Systems Biology. We have now heard back from the three reviewers who agreed to evaluate your study. As you will see below, the reviewers appreciate that the proposed approach addresses a timely topic. However, they raise a series of concerns, which we would ask you to address in a major revision.

Without repeating all the comments listed below, some of the more fundamental issues raised are the following:

- Unique functionalities of your tool should be clarified (Reviewer 2)
- Additional examples of models should be included (simpler or multi-state, Reviewers 1 and 2)

All other issues raised would need to be satisfactorily addressed. Please let me know in case you would like to discuss in further detail any of the comments, I would be happy to schedule a call.

On a more editorial level, we would ask you to address the following point:

- in line with comments from Reviewer 2, we agree that a limitation of the tool is that it requires the costly commercial Matlab software, as opposed to a web-based application that would be more accessible for the community. How can you make the tool more widely available or what is the justification for having it only available in Matlab?

All other issues raised would need to be satisfactorily addressed. Please let me know in case you would like to discuss in further detail any of the comments, I would be happy to schedule a call.

We require:

1) A .docx formatted version of the manuscript text (including legends for main figures, EV figures and tables). Please make sure that the changes are highlighted to be clearly visible. Alternatively you may choose to submit your manuscript as a LaTeX file.

4) A .docx formatted letter INCLUDING the reviewers' reports and your detailed point-by-point responses to their comments. As part of the EMBO Press transparent editorial process, the point-by-point response is part of the Peer Review File (PRF), which will be published alongside your paper.

5) A complete author checklist, which you can download from our author guidelines (<https://www.embopress.org/page/journal/17574684/authorguide#submissionofrevisions>). Please insert information in the checklist that is also reflected in the manuscript. The completed author checklist will also be part of the PRF.

6) Please note that all corresponding authors are required to supply an ORCID ID for their name upon submission of a revised manuscript.

7) It is mandatory to include a 'Data Availability' section after the Materials and Methods. Before submitting your revision, primary datasets produced in this study need to be deposited in an appropriate public database, and the accession numbers and database listed under 'Data Availability'. Please remember to provide a reviewer password if the datasets are not yet public (see <https://www.embopress.org/page/journal/17574684/authorguide#dataavailability>).

In case you have no data that requires deposition in a public database, please state so in this section as follows: "This study includes no data deposited in external repositories". Note that the Data Availability Section is restricted to new primary data that are part of this study.

8) All Materials and Methods need to be described in the main text using our 'Structured Methods' format, which is required for all research articles. According to this format, the Methods section includes a Reagents and Tools Table (listing key reagents, experimental models, software and relevant equipment and including their sources and relevant identifiers) followed by a Methods and Protocols section describing the methods using a step-by-step protocol format. The aim is to facilitate adoption of the methodologies across labs. Please upload the Reagents and Tools table as a separate document when submitting your revised manuscript. More information on how to adhere to this format as well as a downloadable template (.docx) for the Reagents and Tools Table can be found in our author guidelines:
<https://www.embopress.org/page/journal/17444292/authorguide#structuredmethods>

An example of a Method paper with Structured Methods can be found here:
<https://www.embopress.org/doi/10.15252/msb.20178071>.

9) For data quantification: please specify the name of the statistical test used to generate error bars and p-values, the number (n) of independent experiments (specify technical or biological replicates) underlying each data point and the test used to calculate p-values in each figure legend. The figure legends should contain a basic description of n, p-values and the test applied. Graphs must include a description of the bars and the error bars (s.d., s.e.m.). Please provide exact p-values (in either the figure or figure legend).

10) Our journal encourages inclusion of *data citations in the reference list* to directly cite datasets that were re-used and obtained from public databases. Data citations in the article text are distinct from normal bibliographical citations and should directly link to the database records from which the data can be accessed. In the main text, data citations are formatted as follows: "Data ref: Smith et al, 2001" or "Data ref: NCBI Sequence Read Archive PRJNA342805, 2017". In the Reference list, data citations must be labeled with "[DATASET]". A data reference must provide the database name, accession number/identifiers and a resolvable link to the landing page from which the data can be accessed at the end of the reference. Further instructions are available at .

11) We replaced Supplementary Information with Expanded View (EV) Figures and Tables that are collapsible/expandable online. EV Figures should be cited as 'Figure EV1, Figure EV2' etc... in the text and their respective legends should be included in the main text after the legends of regular figures.

- Additional Tables/Datasets should be labeled and referred to as Table EV1, Dataset EV1, etc. Legends should be provided in a separate tab in case of .xls files. Alternatively, the legend can be supplied as a separate text file (README) and zipped together with the Table/Dataset file.

<https://www.embopress.org/page/journal/17574684/authorguide#expandedview>

12) Author contributions: CRediT has replaced the traditional author contributions section because it offers a systematic machine-readable author contributions format that allows for more effective research assessment. Please remove the Authors Contributions from the manuscript and use the free text boxes beneath each contributing author's name in our system to add specific details on the author's contribution. More information is available in our guide to authors.

13) Disclosure statement and competing interests: We updated our journal's competing interests policy in January 2022 and request authors to consider both actual and perceived competing interests. Please review the policy
<https://www.embopress.org/competing-interests> and update your competing interests if necessary.

14) Every published paper now includes a 'Synopsis' to further enhance discoverability. Synopses are displayed on the journal webpage and are freely accessible to all readers. They include a short stand first (maximum of 300 characters, including space) as well as 2-5 one-sentences bullet points that summarizes the paper. Please write the bullet points to summarize the key NEW findings. They should be designed to be complementary to the abstract - i.e. not repeat the same text. We encourage inclusion of key acronyms and quantitative information (maximum of 30 words / bullet point). Please use the passive voice. Please attach these in a separate file or send them by email, we will incorporate them accordingly.

Please note that these would be the final versions and changes during proofing are usually not allowed.

15) As part of the EMBO Publications transparent editorial process initiative (see our policy here: https://www.embopress.org/transparent-process#Review_Process), Molecular Systems Biology will publish online a Peer Review File (PRF) to accompany accepted manuscripts.

In the event of acceptance, this file will be published in conjunction with your paper and will include the anonymous referee reports, your point-by-point response and all pertinent correspondence relating to the manuscript. Let us know whether you

agree with the publication of the PRF and as here, if you want to remove or not any figures from it prior to publication. Please note that the Author checklist will be published at the end of the PRF.

Molecular Systems Biology has a "scooping protection" policy, whereby similar findings that are published by others during review or revision are not a criterion for rejection. Should you decide to submit a revised version, I do ask that you get in touch after three months if you have not completed it, to update us on the status.

Yours sincerely,

Poonam Bheda

Poonam Bheda, PhD
Scientific Editor
Molecular Systems Biology

Reviewer #1:

The paper presents a valuable tool for modeling and simulation of Gene Regulatory Networks through a user friendly app built on Matlab.

Users can build dynamic ODE models of GRNs using the graphical interface. The main components that users define through the GUI include nodes, regulators, and proteases, each with specific properties.

A relevant innovation of the tool is that it incorporates not only transcription-based regulation but also regulation through CRISPRi interactions. Additionally, it allows for the consideration of various degradation kinetics.

The app relies on the Matlab SimBiology Toolbox and is compatible with COPASI solvers.

The app is therefore of relevance for the systems and synthetic biology community, specially for biologists who may not have advanced computational skills.

Next I enumerate a few points that should be considered to improve the manuscript:

1. Although it is commented through the text, a paragraph or a table with a more detailed description of the types of simulation solvers available is needed. It is important to emphasize that the tool supports both deterministic and stochastic simulations. Since it might be not obvious for the user when or why use each of them, some recommendations might be provided.
2. To include a simpler example (for example a toggle switch of transcriptional regulators) providing deterministic and stochastic simulations might be useful, also including the computational times (this could be included in the SI).
3. It would be useful to include computational times of the simulations through the manuscript.
4. The phrase "tool for synthetic biology, aimed to support the entire process of designing, building, and testing synthetic GRNs" could be misleading, as it may suggest that the tool covers the entire DBTL cycle when it is focused specifically on the modeling and simulation of GRNs.
5. I understand that a model created with the app can be automatically translated into SBML. However, it would be helpful to clarify whether it's possible to upload an SBML model to visualize the graph and run simulations within the app. If this is not possible it would be very useful to add this possibility. The compatibility with SBML should be explained more thoroughly in the text, particularly for users who may not be familiar with the SimBiology Toolbox.
6. While an overview of the command-line functionalities is provided in the SI, along with a video tutorial for using the GUI, it would be beneficial to include a concise, self-contained written manual with a simple example to demonstrate the GUI features. This could be incorporated into the SI document or provided alongside the code.

Reviewer #2:

This manuscript presents GRN_modeler, a new computational tool for modeling gene regulatory circuits in systems biology. The authors aim to provide a user-friendly graphical interface to support researchers without extensive programming experience. The tool enables users to construct gene regulatory networks with various interaction types, such as mass-action and Hill kinetics, and perform both ODE and SDE simulations. Additional functionalities include sensitivity analysis, parameter optimization, and fitting. Overall, GRN_modeler has the potential to be a useful tool for the community. However, the authors need to address several concerns regarding its novelty, usability, and applicability.

Major Concerns

1. The modeling and analysis techniques implemented in GRN_modeler are standard in systems biology, and similar functionalities are already available in existing tools. While the case studies presented are interesting, they appear to be achievable using other established methods. The authors should clarify what distinguishes GRN_modeler from existing tools in terms of innovation and unique capabilities.
2. Several existing tools, such as GeneEX, offer GUI-based modeling and simulation of GRNs, and some are web-based, making them more accessible. In contrast, GRN_modeler relies on MATLAB, which is commercial software and may not be readily available to all researchers. The authors should discuss the implications of this limitation and whether alternatives exist for broader accessibility.
3. While the authors claim that the tool supports synthetic circuit design, the provided examples (e.g., actolator and reptolator) merely verify that specified circuits exhibit expected oscillatory behavior. However, a key challenge in circuit design is identifying an unknown network topology that produces a desired dynamic behavior. The manuscript does not address whether GRN_modeler can facilitate such an inverse design process. Additionally, it is unclear how the tool scales to large systems where both circuit topology and parameter space are vast. The authors should clarify these points.
4. The manuscript primarily focuses on oscillatory or excitable circuits like repressilators and feed-forward loops. However, many GRNs exhibit multistability, which is essential in cell fate decisions and synthetic circuit design. How does GRN_modeler handle such systems? Can it identify and analyze multiple stable states effectively?
5. The mathematical modeling section lacks clarity in key areas. For example, it is unclear how the tool models regulatory interactions involving multiple regulators targeting the same gene. The manuscript mentions a two-variable Hill function, but it's unclear to me about its usage in a generic framework. The authors should provide a more detailed explanation.

Minor Points

1. More details on the stochastic simulations are needed, including the choice of noise terms and the type of stochastic analysis performed.
2. In Figure 6, a qualitative comparison between the simulation results and experimental imaging data is necessary. Additionally, it appears that capturing the observed behaviors would require modeling cell growth and PDE-based diffusion, which are beyond the current tool's capabilities.

Reviewer #3:

Review Comments

In this paper, the authors developed new software and investigated oscillatory system dynamics through an integrated approach combining software modeling, circuit design, and experimental validation. They designed novel oscillator families and developed an experimental system for validation.

Some major points are listed below:

1. While the work spans software, circuit design, and experiments, the logical connections between these components are weak. A manuscript excluding any one part would remain coherent, indicating insufficient integrative synthesis.
2. The manuscript omits self-activation, a core mechanism in oscillatory systems (e.g., transcriptional auto-regulation). The authors should clarify whether software limitations preclude its inclusion. Both modeling and experimental designs must address this gap.
3. The method of generating Figure 6D through "rotation" is inappropriate. It requires strong assumptions, which are not always

satisfied in reality. Spatial simulations should be added to accurately represent system dynamics.

4. Some experimental validations for the novel oscillation families (e.g., reptolator and actolator) should be added. The novel oscillation families lack rigorous testing and comparison with established systems.
5. The introduction overly prioritizes technical methods. Expand biological motivation (e.g., natural oscillators like circadian rhythms) to strengthen relevance for the target audience.
6. Results in the novel circuit design section focus on highly symmetrical cases. Include asymmetrical examples to ensure comprehensive analysis and generalizability.
7. Inconsistent node naming between Figure 3 II(a) and II(b). Standardize terminology across sections for clarity.
8. Figure 8 (a) and (b) lack explanation in the text.

Review: “A Tool for Modeling Gene Regulatory Networks (GRN_modeler) and its Applications to Synthetic Biology”

Reviewer #1:

The paper presents a valuable tool for modeling and simulation of Gene Regulatory Networks through a user friendly app built on Matlab. Users can build dynamic ODE models of GRNs using the graphical interface. The main components that users define through the GUI include nodes, regulators, and proteases, each with specific properties. A relevant innovation of the tool is that it incorporates not only transcription-based regulation but also regulation through CRISPRi interactions. Additionally, it allows for the consideration of various degradation kinetics. The app relies on the Matlab SimBiology Toolbox and is compatible with COPASI solvers. The app is therefore of relevance for the systems and synthetic biology community, specially for biologists who may not have advanced computational skills.

We thank the reviewer for the positive feedback and the comments to further improve our work and manuscript. Please find our answers to your points below.

Next I enumerate a few points that should be considered to improve the manuscript:

1. Although it is commented through the text, a paragraph or a table with a more detailed description of the types of simulation solvers available is needed. It is important to emphasize that the tool supports both deterministic and stochastic simulations. Since it might be not obvious for the user when or why use each of them, some recommendations might be provided.

We added a section to the Appendix (Section 1.5, Table S1) that describes the available solvers and provides a brief recommendation to guide users in selecting the most appropriate solver for their specific needs. In the manuscript, we have included the following text to direct readers to this section and emphasize that the application supports both stochastic and deterministic simulations:

The choice between deterministic and stochastic modeling in biological systems depends on the system's scale and the role of intrinsic noise. Deterministic simulations, often using ordinary differential equations (ODEs), are suitable for large systems where molecular fluctuations are negligible, like metabolic networks. In contrast, stochastic simulations (e.g., Gillespie's algorithm) are key for systems with low copy numbers or inherent randomness, such as gene expression or signaling pathways. Hybrid models combine both approaches, using deterministic methods for fast processes and stochastic ones for slow, noise-sensitive reactions. Choosing the right framework is crucial for accurate biological representation. Our software supports both deterministic and stochastic simulations, with available solvers listed in Table S1.

2. To include a simpler example (for example a toggle switch of transcriptional regulators) providing deterministic and stochastic simulations might be useful, also including the computational times (this could be included in the SI). We combined this point with the request of Reviewer 2 who asked for analysis of multistability. We have added an example of the toggle switch to Section 2.7 of the Appendix and analyzed its bistable behavior using both deterministic and stochastic simulations (Figure S7). For the toggle switch, the manuscript has been updated with the following text:

Identifying multistability in GRNs on the example of the toggle switch

In most of our examples so far, the system exhibits either a single stable fixed point (e.g. the FFLs) or a stable periodic orbit (e.g., repressilator and CRISPRlator). However, many GRNs show multistability, a key feature in cell fate decisions and synthetic circuit design. In Appendix Section 2.7, we demonstrate how to identify the steady states of a multistable system using MATLAB's `sbiosteadystate` function, using the toggle switch (Barbier et al, 2020; Perez-Carrasco et al, 2018) – a classic example of a bistable system – as a case study. Figure S7 illustrates how the system transitions between stable states, depending on its initial conditions.

Concerning computational times; for the toggle switch, the deterministic simulations, performed with MATLAB's `ode15s` solver (enhanced by code generation for speed), completed in 0.0015 seconds. Stochastic simulations, conducted with COPASI's `adaptivesa` solver, took 0.0625 seconds. For a more general answer to computational times, please refer to our answer to your next point.

3. It would be useful to include computational times of the simulations through the manuscript.

Since simulation times depend on factors such as the system, applied tolerance values, and solvers, we decided to investigate simulation times using the repressilator family as an example. We varied the number of nodes in the circuit and measured the simulation times with different solvers. The results are summarized in Table S2 and visualized in Figure S3 in the Appendix Section 2.1. This approach allowed us to obtain comparable results for simulation times. Although these times are influenced by factors like the stiffness of the differential equations, they may serve as a useful guide for readers. We added the following text to the manuscript:

In the Appendix, Section 2.1, we show that the scalability of the system is effective, as simulation time increases linearly with system size due to the efficient solvers implemented in MATLAB and COPASI, allowing for the efficient examination of large systems.

4. The phrase “tool for synthetic biology, aimed to support the entire process of designing, building, and testing synthetic GRNs” could be misleading, as it may suggest that the tool covers the entire DBTL cycle when it is focused specifically on the modeling and simulation of GRNs.

We have rephrased the text for clarity. Indeed, our tool specifically focuses on the modeling and simulation of GRNs, helping to provide fast feedback on the dynamical behavior of the GRN, which in turn aids in the development of new circuit designs. We modified the text accordingly:

Our application is an intuitive and valuable tool for synthetic biology, designed specifically to model and simulate synthetic GRNs, providing fast feedback on their dynamical behavior to assist in the development of new circuits.

5. I understand that a model created with the app can be automatically translated into SBML. However, it would be helpful to clarify whether it's possible to upload an SBML model to visualize the graph and run simulations within the app. If this is not possible it would be very useful to add this possibility. The compatibility with SBML should be explained more thoroughly in the text, particularly for users who may not be familiar with the SimBiology Toolbox.

In our original version, while we provided the ability to export and import our models, integrating external models was not yet supported. However, there are indeed many cases where leveraging existing models would be beneficial. To address this, we have now added a functionality to load external models in SBML or SimBiology format and simulate them alongside our system. The interaction between external and new models is facilitated through shared

species. Furthermore, also in response to the reviewer's next comment, we have created a GUI manual in Appendix Section 1.4, where we describe this new functionality in detail under the "Adding External Models" section.

6. While an overview of the command-line functionalities is provided in the SI, along with a video tutorial for using the GUI, it would be beneficial to include a concise, self-contained written manual with a simple example to demonstrate the GUI features. This could be incorporated into the SI document or provided alongside the code.

In response to the reviewer's comment, we have created a concise, self-contained written manual for the GUI in the Appendix Section 1.4. This manual is based on our video tutorial and includes additional notes and examples to help users effectively navigate and utilize the GUI features.

Reviewer #2:

This manuscript presents GRN_modeler, a new computational tool for modeling gene regulatory circuits in systems biology. The authors aim to provide a user-friendly graphical interface to support researchers without extensive programming experience. The tool enables users to construct gene regulatory networks with various interaction types, such as mass-action and Hill kinetics, and perform both ODE and SDE simulations. Additional functionalities include sensitivity analysis, parameter optimization, and fitting. Overall, GRN_modeler has the potential to be a useful tool for the community. However, the authors need to address several concerns regarding its novelty, usability, and applicability.

We thank the reviewer for the positive feedback and the comments to further improve our work and manuscript. Please find our answers to your points below. We sincerely hope we could address your concerns regarding novelty, usability and applicability.

Major Concerns

1. The modeling and analysis techniques implemented in GRN_modeler are standard in systems biology, and similar functionalities are already available in existing tools. While the case studies presented are interesting, they appear to be achievable using other established methods. The authors should clarify what distinguishes GRN_modeler from existing tools in terms of innovation and unique capabilities.

While GRN_modeler uses standard techniques, it stands out due to its dual functionality: an intuitive GUI for accessibility and a command-line interface for scalability and advanced analysis. Additionally, (in its new version) it enables spatial simulations of gene regulatory networks, specifically simulation on growing colonies and lawns of cells, which are not commonly available in other tools. In a new paragraph we highlight in the manuscript, that GRN_modeler combines ease of use, flexibility, and advanced features, making it a valuable addition to existing GRN modeling tools:

Although many excellent tools exist for simulating GRNs – such as LOICA (Vidal et al, 2022), which offers command-line functionalities, GeneEX (Kohar et al, 2020), a user-friendly web-based platform, or LimbNET (preprint: Matyjaskiewicz & Sharpe, 2024), for spatial simulations in limb development – our application stands out in several important ways. It provides a GUI, making it accessible to users without programming experience, while also supporting command-line use. This additional functionality is particularly valuable for building larger systems or performing extended analyses not supported by the GUI, significantly increasing the flexibility of the tool.

Additionally, our GUI can generate scripts to replicate the circuit designs, further enhancing the convenience and power of the command-line interface. Beyond circuit design and simulation, our tool also enables spatial simulations, including growing colonies and cell lawns. We demonstrate this capability through examples such as non-autonomous period-2 oscillations and Turing pattern formation. We believe that this combination of usability, flexibility, and advanced features makes our application a valuable addition to the current suite of GRN modeling tools.

2. Several existing tools, such as GeneEX, offer GUI-based modeling and simulation of GRNs, and some are web-based, making them more accessible.

We appreciate the reviewer for pointing out the interactive web-app GeneEX, which we had overlooked. While we mentioned several simulation software in the introduction, such as XPP/XPPAUT, KinTek Explorer, TABASCO, COPASI, Bioscrape, libRoadRunner, SBMLsimulator, SimBiology, txttools, iBioSim, TX-TLsim, Metabolic Tinker, WebCM, Flapjack, and LOICA, GeneEX is indeed particularly relevant to our work. We now cite GeneEX and

discuss it in the paragraph that we wrote to address the reviewer’s first point concerning the uniqueness of our tool. Please, see above.

In contrast, GRN_modeler relies on MATLAB, which is commercial software and may not be readily available to all researchers. The authors should discuss the implications of this limitation and whether alternatives exist for broader accessibility.

We have added a dedicated paragraph to the manuscript discussing the choice of MATLAB for the implementation of our application. In this section, we address the rationale behind selecting MATLAB, as well as the potential limitations due to its commercial nature and restricted availability. While MATLAB offers robust numerical capabilities and a well-supported environment for scientific computing—particularly beneficial for stiff solvers, parallelization and matrix-based operations—we acknowledge that its licensing requirements may limit accessibility for some users. We hope this clarification helps contextualize our decision. The added paragraph reads as follows:

Our decision to use MATLAB in this study reflects a careful consideration of its strengths and limitations. While we acknowledge that MATLAB is a commercial platform, its multi-platform compatibility (Windows, macOS, Linux) and widespread adoption in academia ensure accessibility for our target audience. The software provides a robust environment for numerical simulations, leveraging highly optimized solvers such as ode15s for stiff systems and SUN-DIALS CVODE for large-scale differential equations. Additionally, the sbioccelerate function offers significant speedup, which is especially valuable for computationally intensive tasks. In the spatial simulations, we also take advantage of MATLAB’s highly optimized matrix operations – such as matrix-vector multiplication, LU decomposition, and solving linear systems. These tools enable efficient handling of computationally intensive tasks, complemented by the Parallel Computing Toolbox for accelerated parameter sweeps. Although MATLAB’s licensing cost may pose a barrier for some users, its long-term stability and backward compatibility reduce maintenance burdens compared to open-source alternatives, which often require frequent updates to maintain functionality.

Furthermore, the *SimBiology toolbox* in MATLAB provides numerous powerful features, including an intuitive graphical interface, which we have extended by integrating it with the free *COPASI* software for added flexibility. In addition to its graphical user interface, our application offers programmatic tools that are particularly beneficial for advanced users. These users can build models programmatically and perform further analyses, such as finding fixed points, evaluating the Jacobian for system stability, or running spatial simulations for growing colonies, leveraging MATLAB’s extensive capabilities. In summary, the choice of MATLAB allows us to provide a reliable, feature-rich platform with high computational performance and long-term stability.

3. While the authors claim that the tool supports synthetic circuit design, the provided examples (e.g., actolator and reptolator) merely verify that specified circuits exhibit expected oscillatory behavior. However, a key challenge in circuit design is identifying an unknown network topology that produces a desired dynamic behavior. The manuscript does not address whether GRN_modeler can facilitate such an inverse design process.

The reviewer is correct in pointing out that we did not sufficiently emphasize the role of the user in the design process. Indeed, the application alone cannot perform the inverse engineering task. However, we believe that our examples using the actolator and reptolator effectively illustrate how the application significantly accelerates the design process in reverse engineering. In these examples, our goal was to design oscillators that can oscillate with an even number of nodes, which directly addresses the “key challenge” highlighted by the reviewer. However, this process is not fully automated, the application serves primarily to alleviate the complexities of model building and simulation running. The user still plays a central role in generating new ideas based on the simulation results of

different circuit designs. We have made an effort to clarify this concept in the manuscript, particularly by modifying Figure 1 to better reflect the role of human creativity in the design process. Additionally, we have expanded and revised the description of the final step within our application, incorporating these insights.

In the final step, simulations are run to observe the system’s dynamic behavior. The goal of our application is to accelerate circuit design and provide an easy way to visualize potential outputs. If the circuit does not exhibit the expected behavior during simulations, the design can be revisited in step II, allowing for adjustments before repeating the process. In this step, human creativity plays a crucial role. While the application assists in predicting and visualizing circuit behavior, it is the user’s knowledge and intuition that guide the necessary modifications to the design.

The current version of our application does not support inverse design, but it has the potential to be extended in this direction in the future. Accordingly, we have updated the Discussion section to reflect this limitation as follows:

Another limitation is that the user must manually define the circuits to be tested, as the current version of GRN-modeler does not support automatic generation of network designs that produce a desired behavior. Addressing this challenge requires a different set of approaches, and various strategies have been explored in the literature, including dedicated design toolboxes (Otero-Muras et al, 2016), brute-force screening methods (Cotterell & Sharpe, 2010), and machine learning-based frameworks (Palacios et al, 2025).

Additionally, it is unclear how the tool scales to large systems where both circuit topology and parameter space are vast. The authors should clarify these points.

The scalability of the system raises two key questions: 1) How quickly can the system be built? and 2) How much time does it take to simulate the system? Regarding the first question, we demonstrated with the example of our oscillators that the application’s programmatic capabilities make it easy to generate larger circuits, particularly when they feature repeating elements or a logical structure. We calculated the time period and amplitude of these oscillators for up to 20 nodes. While extending to higher node numbers would be straightforward, for synthetic biology the practical relevance of such large circuits is limited, as we are currently not able to build such complex networks. As for simulation time, in response to the first reviewer’s request, we analyzed the impact of system size (i.e., the number of nodes) using different solvers. These results are summarized in Appendix Section 2.1, Table S2 and illustrated in Figure S3. Based on these results, we observe that the simulation time scales efficiently, particularly in the deterministic models, thanks to the effectiveness of the solvers, making it possible to build and simulate large systems. We added the following paragraph to the manuscript:

In summary, by creating these novel oscillatory families, we have demonstrated how to use GRN_modeler to go from a desired behaviour (oscillations with even numbers of nodes) to circuit design. Although this is not an automatic process, the tool accelerates circuit design by revealing the system’s dynamic behavior. It is also worth noting the advantage of the command-line functionalities, which allow for the modular construction of large circuits. This capability enabled us to easily test the behavior of circuits with a large number of nodes. In the Appendix, Section 2.1, we show that the scalability of the system is effective, as simulation time increases linearly with system size due to the efficient solvers implemented in MATLAB and COPASI, allowing for the efficient examination of large systems.

Given the vastness of the parameter space, the GUI currently supports parameter scanning for only one parameter at a time, while keeping all others fixed. For circuits in which only a small fraction of the parameter space yields the desired behavior, identifying suitable parameter combinations through repeated manual scans can be highly inefficient. To address this, the SimBiology Toolbox offers both local and global sensitivity analysis, which can be parallelized

to accelerate these computationally expensive simulations. By identifying the most influential parameters, users can then apply our tool’s parameter scan functionality more effectively as we demonstrated in Figure 4. However, we acknowledge that this type of analysis is generally non-trivial and may require prior modeling experience.

4. The manuscript primarily focuses on oscillatory or excitable circuits like repressilators and feed-forward loops. However, many GRNs exhibit multistability, which is essential in cell fate decisions and synthetic circuit design. How does GRN_modeler handle such systems? Can it identify and analyze multiple stable states effectively?

We agree with the reviewer that multistability is a key feature of many GRNs. Since SimBiology includes a built-in function, `sbiosteadystate`, to find steady states, integrating this functionality into our application was straightforward. This addition enhances the user’s ability to explore and better understand the system’s behavior. Given Reviewer 1’s interest in including the toggle switch as an example, we chose to demonstrate this functionality using the bistable toggle switch model. We added this example to the Appendix Section 2.7, Figure S7. While this approach does not fully capture the system’s complete dynamics – for instance, identifying unstable periodic orbits would require additional methods – it provides an accessible way to uncover the system’s fundamental properties. We have extended the manuscript with the following paragraph:

Furthermore, we included a brief explanation of this functionality in the “Parameter Scan” section of the new GUI manual in Appendix Section 1.4.

5. The mathematical modeling section lacks clarity in key areas. For example, it is unclear how the tool models regulatory interactions involving multiple regulators targeting the same gene. The manuscript mentions a two-variable Hill function, but it’s unclear to me about its usage in a generic framework. The authors should provide a more detailed explanation.

We thank the reviewer for pointing out this lack of information. To clarify this, we have added Section 1.2 to the Appendix, where we provide a comprehensive explanation of how regulatory interactions are represented and implemented in our tool.

In summary, GRN_modeler supports both standard and complex regulatory cases. For standard cases—where regulation can be expressed as a product of Hill functions—the GUI can be used directly without modifications. In more complex cases, where multi-input regulatory interactions require special multivariable functions, users can modify the corresponding SimBiology rules to describe these interactions explicitly. This flexibility allows our tool to support hierarchical regulatory chains and multi-input regulations of arbitrary depth using any MATLAB-compatible function. We direct the reader to this section of the manuscript with the following statement:

In Appendix Figure S13, we demonstrate how this two-input node can be handled within GRN_modeler, **while Appendix Section 1.2 summarizes the different ways regulatory interactions can be interpreted within our tool.**

Minor Points

1. More details on the stochastic simulations are needed, including the choice of noise terms and the type of stochastic analysis performed.

We have expanded the description of the stochastic simulations and available solvers (Appendix Section 1.5). In these simulations, noise arises intrinsically from molecular discreteness and reaction probabilities, governed by the Chemical Master Equation (CME). Discrete reaction events are sampled probabilistically based on reaction propensities (Gillespie-type noise), without additional extrinsic noise terms such as additive Wiener processes. However,

COPASI allows for additional noise sources, including Langevin-like fluctuations, which can be introduced via rate rules. For our stochastic simulations, we employed COPASI’s Adaptive SSA solver, which dynamically transitions between the Gillespie Direct Method and the τ -leap approximation depending on system state. This approach ensures both accuracy and computational efficiency, particularly for larger systems. To provide further clarity, we have also included a brief description of additional stochastic solvers in Appendix Section 1.5. The following text has been added to the manuscript:

2. In Figure 6, a qualitative comparison between the simulation results and experimental imaging data is necessary.

The reviewer is correct that Figure 6 presents only a qualitative comparison between the simulated and experimental data. However, this aligns with our objective in this section. Our aim was to demonstrate the application’s ability to generate insightful and useful qualitative predictions. To illustrate this, we first presented the predicted qualitative outcomes, followed by the corresponding experimental results. Achieving quantitative agreement using the default parameters of the Elowitz model was not the goal in this case, as such an approach would require extensive parameter fitting, which would shift the focus from prediction to mere curve fitting. Instead, we provide an example of parameter fitting and quantitative comparison in Figure 7, specifically for our light-inducible system. To give a qualitative comparison between the simulation results and experimental imaging data we added the following sentence to the manuscript:

Additionally, it appears that capturing the observed behaviors would require modeling cell growth and PDE-based diffusion, which are beyond the current tool’s capabilities.

It is true that modeling cell growth and PDE-based diffusion was beyond the capabilities of the previous version of our application. However, in response to the reviewer’s comment regarding uniqueness and also addressing a related comment from Reviewer 3, we have now implemented spatial simulation capabilities for both growing colonies and cell lawns. These new features are demonstrated in the manuscript with examples of period-2 oscillations and Turing pattern formation, presented in a newly added section titled “Spatial Simulations” (including Figure 10). Further numerical details on the spatial simulations are also provided in the Appendix (Section 1.3.1).

Reviewer #3:

In this paper, the authors developed new software and investigated oscillatory system dynamics through an integrated approach combining software modeling, circuit design, and experimental validation. They designed novel oscillator families and developed an experimental system for validation.

We thank the reviewer for the comments to further improve our work and manuscript. Please find our answers to your points below.

Some major points are listed below:

1. While the work spans software, circuit design, and experiments, the logical connections between these components are weak. A manuscript excluding any one part would remain coherent, indicating insufficient integrative synthesis.

We agree with the reviewer’s comment that the description of the GRN_modeler tool, the new circuits we explored, and the experimental example based on the light biosensor could each serve as the foundation for separate studies. However, we wanted to illustrate the usefulness of our tool with concrete examples. Similarly to a textbook, where examples enhance the applicability of a theory, our novel and practical examples demonstrate how the tool can be applied in specific cases. We hope that the application of our software provides a sufficient unifying thread, and despite the distinct sections, the structure of the paper remains clear and cohesive. It is true that the manuscript would still hold value even if any of these examples were removed, but we believe this also presents an advantage. Specifically, if an experimentalist is only interested in the light biosensor, they can benefit from the paper without needing to fully engage with every aspect of GRN_modeler. This approach allows readers interested only in specific examples to derive value from the study, which we hope will further broaden its impact.

2. The manuscript omits self-activation, a core mechanism in oscillatory systems (e.g., transcriptional auto-regulation). The authors should clarify whether software limitations preclude its inclusion. Both modeling and experimental designs must address this gap.

It is indeed possible to implement self-regulating nodes in the application. To demonstrate this, we have added two new examples: the Goodwin oscillator – a self-repressing node – and the Stricker (or dual-feedback) oscillator (Appendix Section 2.6). Since the representation of self-repressing and self-activating nodes in the application is non-trivial (Appendix Figure S5, Figure S6), these examples serve as a valuable guide for users. On the other hand, self-activation has already been experimentally implemented in other studies (e. g. (Stricker et al, 2008)), and showing it again experimentally is outside the scope of this manuscript. In this manuscript, we just wanted to provide one example to illustrate the integration between the app and wet-lab. We have added the following text to the main manuscript to direct readers to these examples:

Finally, to demonstrate the implementation of self-regulating nodes in GRN_modeler and provide additional examples, we implemented the model of two more oscillators, namely the Goodwin oscillator (Goodwin, 1963) (Figure S5) and the dual-feedback oscillator (Stricker et al, 2008) (Figure S6) in the Appendix, Section 2.6.

3. The method of generating Figure 6D through "rotation" is inappropriate. It requires strong assumptions, which are not always satisfied in reality. Spatial simulations should be added to accurately represent system dynamics.

We appreciate the reviewer’s challenging yet insightful comment. In response, we have implemented spatial simulation capabilities for both growing colonies and cell lawns. We demonstrate these new features in the manuscript with examples of period-2 oscillations and Turing pattern formation, presented in a newly added section titled "Spatial

Simulations” (including Figure 10). Additionally, we provide further numerical details on the spatial simulations in the Appendix (Section 1.3.1). Although the pseudo-two-dimensional image (Figure 6d) relies on strong assumptions, it may still be helpful to users, as it provides a more intuitive visual impression of the colony’s appearance than time series alone—and it can be generated quickly.

4. Some experimental validations for the novel oscillation families (e.g., reptolator and actolator) should be added. The novel oscillation families lack rigorous testing and comparison with established systems.

We agree that building and experimentally testing these circuits would be valuable. However, building and fine-tuning these circuits requires extensive experimental efforts, typically spanning several months to years. This is beyond the scope of this primarily theoretical study. Our work provides a systematic framework for designing and analyzing these circuits, offering a foundation for future experimental investigations. We hope that this study will inspire further research, leading to rigorous testing and comparison with established systems.

5. The introduction overly prioritizes technical methods. Expand biological motivation (e.g., natural oscillators like circadian rhythms) to strengthen relevance for the target audience.

To expand the biological relevance, we have added the following paragraph to the introduction:

Among the most well-studied designs are feed-forward loops (FFLs), toggle switches, and oscillators. Feed-forward loops involves a gene regulating another gene both directly and indirectly through another regulator. This architecture is commonly found in most living organisms (Weldemichael et al, 2022; Mangan & Alon, 2003; Alon, 2007) and can filter noise, produce sign-sensitive delays, create pulse-like responses or produce spatial stripe patterns (Lee et al, 2002; Milo et al, 2002; Alon, 2007; Schaerli et al, 2014; Santos-Moreno et al, 2020). The toggle switch, composed of two mutually repressive genes, enables bistability – cells can stably exist in one of two states – which is crucial in biological systems like the cell fate decisions (Gardner et al, 2000; Barbier et al, 2020; Perez-Carrasco et al, 2018; Saka & Smith, 2007; Briscoe & Small, 2015; Clark, 2017). Genetic oscillators (Li & Yang, 2018), such as the repressilator (Elowitz & Leibler, 2000; Potvin-Trottier et al, 2016) generate cyclic gene expression, emulating a variety of physiological phenomena, from circadian rhythms, to cell cycle dynamics or spatiotemporal pattern formation during organisms’ development (Saini et al, 2019; Poon, 2016; Gomez et al, 2008; Tsiairis & Aulehla, 2016). These synthetic circuits not only deepen our understanding of natural regulatory networks but also enable applications in biotechnology and therapeutic systems (Xie & Fussenegger, 2018; Kumar & Hasty, 2023).

6. Results in the novel circuit design section focus on highly symmetrical cases. Include asymmetrical examples to ensure comprehensive analysis and generalizability.

We agree with the reviewer that the asymmetry in these circuits is a fascinating phenomenon, which we explored in Section 2.8.2 of the Appendix. In the “acrelator” family, replacing repressive interactions with activations introduces this asymmetry, as certain nodes activate simultaneously due to activation interactions, reducing the number of regulatory steps. For instance, the “8-3” circuit illustrates how an even-numbered circuit with activations can behave like a smaller repressilator. The asymmetry is evident in the differing node sizes and synchronized activations, resulting in varying duty cycles—nodes later in the activation chain have longer duty cycles. This feature enables customization of the circuit’s oscillatory behavior and can help reduce cellular burden by selecting nodes with shorter duty cycles for specific applications. Additionally, in response to the reviewer’s previous comment (point 2), we also examined the Stricker oscillator (Figure S6), which is another asymmetric system.

7. Inconsistent node naming between Figure 3 II(a) and II(b). Standardize terminology across sections for clarity.

We have updated the node names in Figure II(b) for consistency. We initially followed the nomenclature from Tomazou *et al.* (Tomazou et al, 2018), however, the revised naming scheme ensures greater consistency throughout our manuscript.

8. Figure 8 (a) and (b) lack explanation in the text.

We thank the reviewer for bringing this to our attention. We added reference and explanation for Figure 8 (a) and (b) in the text:

Using the model with the fitted parameters, we simulated the patterns generated by the biosensor in response to light, in absence (Figure 8a) or presence of L-arabinose (Figure 8b). The simulations predicted that the biosensor would form rings when exposed to light pulses and L-arabinose concentrations exceeding $> 10^{-4}\%$. Remarkably, these rings would be in anti-phase compared to the rings produced in the absence of L-arabinose (that is, dark rings would replace bright rings, and vice versa; Figure 8a-c).

Editorial questions

- Unique functionalities of your tool should be clarified (Reviewer 2)

We thank the editor for raising this point. Compared to existing tools, GRN_modeler stands out by combining an intuitive GUI for ease of use with a command-line interface for scalability and advanced analysis. Its unique capability to perform spatial simulations of gene regulatory networks, such as growing colonies and simulations on lawns of cells, further distinguishes it from existing tools. For more details, we have addressed these points in response to Reviewer 2's first question.

- Additional examples of models should be included (simpler or multi-state, Reviewers 1 and 2)

In response to the reviewers' suggestions, we have incorporated the following examples: stability analysis of the toggle switch, a self-regulating oscillator, the Goodwin oscillator, and the asymmetric, self-regulating Stricker oscillator. Additionally, we expanded the application to support spatial simulations, including a light-inducible oscillator exhibiting period-2 oscillations and the Stricker oscillator generating Turing patterns.

- in line with comments from Reviewer 2, we agree that a limitation of the tool is that it requires the costly commercial Matlab software, as opposed to a web-based application that would be more accessible for the community. How can you make the tool more widely available or what is the justification for having it only available in Matlab?

We appreciate the editor's comment regarding the use of MATLAB in GRN_modeler. As noted, MATLAB is a commercial platform, which may limit accessibility for some researchers. However, MATLAB offers robust capabilities for numerical simulations, optimized solvers, and matrix operations, which are crucial for handling complex, computationally intensive tasks. These features ensure the tool's reliability and efficiency in modeling GRNs, particularly for stiff systems and spatial simulations.

We discuss the choice of MATLAB, its strengths, and potential limitations in the manuscript, where we also mention how the integration with free tools like COPASI adds flexibility. Although MATLAB's licensing cost may pose a barrier for some users, its long-term stability and backward compatibility reduce maintenance burdens compared to open-source alternatives, which often require frequent updates to maintain functionality. For more detailed insights into the rationale behind using MATLAB and how it benefits the tool, we direct the editor to the dedicated paragraph in the manuscript that addresses these concerns, as well as to our answer to the second question of Reviewer 2.

11th Aug 2025

Manuscript Number: MSB-2025-12880R

Title: A Tool for Modeling Gene Regulatory Networks (GRN_modeler) and its Applications to Synthetic Biology

Dear Prof Schaerli,

Thank you for the submission of your revised manuscript to Molecular Systems Biology. We have now received the enclosed reports from the referees that were asked to re-assess it. As you will see the reviewers are now globally supportive and I am pleased to inform you that we will be able to accept your manuscript pending the following final amendments:

- 1) In the main manuscript file, please include keywords to max. 5.
- 2) For the Data availability section, it is not necessary to include information on plasmid sequences available via Addgene (we would suggest including this information directly in the Methods). The Source Data information is also not necessary, a separate statement/link will be added for your Source Data by Production. In addition, please remove the "Code Availability" section and just include this information in "Data Availability", formatted according to the example below:
"The datasets and computer code produced in this study are available in the following databases:
- Chip-Seq data: Gene Expression Omnibus GSE46748 (<https://www.ncbi.nlm.nih.gov/geo/query/acc.cgi?acc=GSE46748>)
- Modeling computer scripts: GitHub (<https://github.com/SysBioChalmers/GECKO/releases/tag/v1.0>)
- [data type]: [full name of the resource] [accession number/identifier] ([doi or URL or identifiers.org/DATABASE:ACCESSION])"
- 3) Please rename "Competing Interests" to "Disclosure and competing interests statement". We updated our journal's competing interests policy in January 2022 and request authors to consider both actual and perceived competing interests. Please review the policy <https://www.embopress.org/competing-interests> and update your competing interests if necessary.
- 4) Please remove the "Supplementary Material" section of the manuscript, and provide this information elsewhere in the Methods as appropriate.
- 5) Please place individual sections of the manuscript in the following order: Title page - Abstract & Keywords - Introduction - Results - Discussion - Methods - Data Availability - Acknowledgements - Disclosure and Competing Interests Statement - References - Figure Legends - Expanded View Figure Legends.
- 6) For the figures and figure legends, please take care of the following:
 - Please remove all figures from main manuscript file and leave only main figure legends placed after the references.
 - All figure callouts should be listed sequentially in the main manuscript. In addition, callouts are missing for Fig. 2A-F, 3B-D, 4C-D, 10B-C, Appendix Figure S1-S3, S10-S12, Appendix Table S2
 - In a routine figure check, we observed that images were reused between Figure 6E and Appendix Fig 17A, as well as between Figure 8D&E and Appendix Fig 17A&B. This is acceptable, however it should be stated clearly in each the legend.
- 7) The movies should be called out as Movie EV1-EV8, and the legends should be zipped with each movie file. Each movie should be uploaded as an individual zip folder to our manuscript submission system.
- 8) In the Appendix file, the title page should contain "Appendix for + manuscript title" and a table of contents with the page numbers for the listed items, which should also include the Appendix Figures and Tables. The nomenclature should be Appendix Figure Sx and Appendix Table Sx throughout manuscript and the Appendix itself.
- 9) Please provide the synopsis image as a high-resolution JPEG or TIFF file with the dimensions 550 pixels wide x (300-600) pixels high. Please also remove the synopsis image from the manuscript file.
- 10) Source Data: Please upload the Source Data checklist as a Related Manuscript File (and not in the zipped Source Data folder). Source Data should be organized as a single source data file (zipped) per figure for main figures (all EV and/or Appendix figure Source Data can be included in a single folder), with the panels clearly visible in the folder structure instead of a single excel file for all Source Data. e.g. all the Source data files for figure 1 need to be saved in a single folder and this needs to be zipped and then uploaded as "SD figure 1.zip" file.
- 11) As part of the EMBO Publications transparent editorial process initiative (see our policy here: https://www.embopress.org/transparent-process#Review_Process), Molecular Systems Biology will publish online a Peer Review File (PRF) to accompany accepted manuscripts. This file will be published in conjunction with your paper and will include the anonymous referee reports, your point-by-point response and all pertinent correspondence relating to the manuscript. Let us know whether you agree with the publication of the PRF and as here, if you want to remove or not any figures from it prior to publication. Please note that the Authors checklist will be published at the end of the PRF.
- 12) After your paper is published, we may promote it on social media. If you have any handles or hashtags for Bluesky you would like included, please let us know.
- 13) Please provide a point-by-point letter INCLUDING my comments and your detailed responses (as Word file).

I look forward to reading a new revised version of your manuscript as soon as possible.

Yours sincerely,

Poonam Bheda, PhD
Scientific Editor

Reviewer #1:

The authors have satisfactorily addressed all the comments raised during the revision.

Reviewer #2:

The revised manuscript has properly addressed my comments and questions. I recommend publication of this manuscript.

I checked 3rd reviewers' questions and the responses from the authors in the revision.

I believe the authors have sufficiently addressed the reviewers' concerns. In the revision, they added new examples of gene circuit modeling and spatial modeling to directly respond to the critiques. While no validation was provided for the predicted oscillatory circuits, I agree with the authors that such validation is beyond the scope of the current study.

1) In the main manuscript file, please include keywords to max. 5.

Added keywords: simulation tool, gene regulatory networks, light biosensor, synthetic biology, optogenetics

2) For the Data availability section, it is not necessary to include information on plasmid sequences available via Addgene (we would suggest including this information directly in the Methods). The Source Data information is also not necessary, a separate statement/link will be added for your Source Data by Production. In addition, please remove the "Code Availability" section and just include this information in "Data Availability", formatted according to the example below:

"The datasets and computer code produced in this study are available in the following databases:

- Chip-Seq data: Gene Expression Omnibus GSE46748

(<https://www.ncbi.nlm.nih.gov/geo/query/acc.cgi?acc=GSE46748>)

- Modeling computer scripts: GitHub

(<https://github.com/SysBioChalmers/GECKO/releases/tag/v1.0>)

- [data type]: [full name of the resource] [accession number/identifier] ([doi or URL or identifiers.org/DATABASE:ACCESSION])"

Plasmids information was moved to Methods section and Data Availability was updated. Source Data and Code Availability sections were deleted.

3) Please rename "Competing Interests" to "Disclosure and competing interests statement". We updated our journal's competing interests policy in January 2022 and request authors to consider both actual and perceived competing interests. Please review the policy <https://www.embopress.org/competing-interests> and update your competing interests if necessary.

Renaming done, no update on competing interests was necessary.

4) Please remove the "Supplementary Material" section of the manuscript, and provide this information elsewhere in the Methods as appropriate.

We removed the "Supplementary Material" section and moved the content to the Methods section.

5) Please place individual sections of the manuscript in the following order: Title page - Abstract & Keywords - Introduction - Results - Discussion - Methods - Data Availability - Acknowledgements - Disclosure and Competing Interests Statement - References - Figure Legends - Expanded View Figure Legends.

Order updated.

6) For the figures and figure legends, please take care of the following:

- Please remove all figures from main manuscript file and leave only main figure legends placed after the references.

Done.

- All figure callouts should be listed sequentially in the main manuscript. In addition, callouts

are missing for Fig. 2A-F, 3B-D, 4C-D, 10B-C, Appendix Figure S1-S3, S10-S12, Appendix Table S2

Done.

- In a routine figure check, we observed that images were reused between Figure 6E and Appendix Fig 17A, as well as between Figure 8D&E and Appendix Fig 17A&B. This is acceptable, however it should be stated clearly in each the legend.

Thank you for checking the figures. We added a statement clarifying the reused images in both Figures 6, Figure 8 and Appendix Figure S17 captions.

7) The movies should be called out as Movie EV1-EV8, and the legends should be zipped with each movie file. Each movie should be uploaded as an individual zip folder to our manuscript submission system.

We prepared an individual zip for each (movie + subtitle) and added a callout for the movies in the Methods section.

8) In the Appendix file, the title page should contain "Appendix for + manuscript title" and a table of contents with the page numbers for the listed items, which should also include the Appendix Figures and Tables. The nomenclature should be Appendix Figure Sx and Appendix Table Sx throughout manuscript and the Appendix itself.

Done.

9) Please provide the synopsis image as a high-resolution JPEG or TIFF file with the dimensions 550 pixels wide x (300-600) pixels high. Please also remove the synopsis image from the manuscript file.

We provide the TIFF file with the correct dimensions and removed it from the manuscript.

10) Source Data: Please upload the Source Data checklist as a Related Manuscript File (and not in the zipped Source Data folder). Source Data should be organized as a single source data file (zipped) per figure for main figures (all EV and/or Appendix figure Source Data can be included in a single folder), with the panels clearly visible in the folder structure instead of a single excel file for all Source Data. e.g. all the Source data files for figure 1 need to be saved in a single folder and this needs to be zipped and then uploaded as "SD figure 1.zip" file.

Done

11) As part of the EMBO Publications transparent editorial process initiative (see our policy here: https://www.embopress.org/transparent-process#Review_Process), Molecular Systems Biology will publish online a Peer Review File (PRF) to accompany accepted manuscripts. This file will be published in conjunction with your paper and will include the anonymous referee reports, your point-by-point response and all pertinent correspondence relating to the manuscript. Let us know whether you agree with the publication of the PRF and as here, if you want to remove or not any figures from it prior to publication. Please note that the Authors checklist will be published at the end of the PRF.

We agree with the peer review file as it is.

12) After your paper is published, we may promote it on social media. If you have any handles or hashtags for Bluesky you would like included, please let us know.

@yschaerli.bsky.social

@dmf-unil.bsky.social

@fbm-unil.bsky.social

@unil.bsky.social

13) Please provide a point-by-point letter INCLUDING my comments and your detailed responses (as Word file).

Done

1st Sep 2025

Manuscript number: MSB-2025-12880RR

Title: A Tool for Modeling Gene Regulatory Networks (GRN_modeler) and its Applications to Synthetic Biology

Dear Prof Schaerli,

Thank you again for sending us your revised manuscript. I am pleased to inform you that your paper has been accepted for publication in Molecular Systems Biology.

Yours sincerely,

Sincerely,

Poonam Bheda, PhD
Scientific Editor
Molecular Systems Biology
